# Practical Large-Scale Linear Programming using Primal-Dual Hybrid Gradient

**David Applegate**
Google Research
dapplegate@google.com

**Mateo Díaz**
California Institute of Technology*
mateodd@caltech.edu

**Oliver Hinder**
Google Research
University of Pittsburgh
ohinder@pitt.edu

**Haihao Lu**
University of Chicago†
haihao.lu@chicagobooth.edu

**Miles Lubin**
Google Research
mlubin@google.com

**Brendan O'Donoghue**
DeepMind
bodonoghue@deepmind.com

**Warren Schudy**
Google Research
wschudy@google.com

## Abstract

We present PDLP, a practical first-order method for linear programming (LP) that can solve to the high levels of accuracy that are expected in traditional LP applications. In addition, it can scale to very large problems because its core operation is matrix-vector multiplications. PDLP is derived by applying the primal-dual hybrid gradient (PDHG) method, popularized by Chambolle and Pock (2011), to a saddle-point formulation of LP. PDLP enhances PDHG for LP by combining several new techniques with older tricks from the literature; the enhancements include diagonal preconditioning, presolving, adaptive step sizes, and adaptive restarting. PDLP improves the state of the art for first-order methods applied to LP. We compare PDLP with SCS, an ADMM-based solver, on a set of 383 LP instances derived from MIPLIB 2017. With a target of $10^{-8}$ relative accuracy and 1 hour time limit, PDLP achieves a 6.3x reduction in the geometric mean of solve times and a 4.6x reduction in the number of instances unsolved (from 227 to 49). Furthermore, we highlight standard benchmark instances and a large-scale application (PageRank) where our open-source prototype of PDLP, written in Julia, outperforms a commercial LP solver.

## 1 Introduction

First-order methods (FOMs), which use gradient and not Hessian information, are now applied as standard practice in many areas of optimization [12]. A known weakness of FOMs is the *tailing-off* effect, where FOMs quickly find moderately accurate solutions, but progress towards an optimal solution slows down over time. While moderately accurate solutions are often sufficient for large machine learning applications, other applications traditionally demand higher precision. One such area is Linear Programming (LP), the focus of this work.

LP is a fundamental class of optimization problems in applied mathematics, operations research, and computer science with a huge range of applications, including mixed-integer programming, scheduling, network flow, chip design, budget allocation, and many others [17, 22, 65, 69]. Software

---

*Part of the work was done at Cornell and Google Research.
†Part of the work was done at Google Research.

35th Conference on Neural Information Processing Systems (NeurIPS 2021).

for solving LP problems, called *LP solvers*, originated in the earliest days of computing, predating the invention of operating systems [55]. The state-of-the-art methods for LP, namely Dantzig's simplex method [22, 23] and interior-point (or barrier) methods [51], are quite mature and reliable at delivering highly accurate solutions. These widely successful methods have left little room for FOMs to make inroads. Furthermore, practitioners who use LP solvers are not accustomed to reasoning about the trade-off between accuracy and computing times typically intrinsic to FOMs.

In this paper, we provide evidence that, if properly enhanced, FOMs can obtain high quality solutions to LP problems quickly. Indeed, there's reason to expect this, as authors have developed FOMs for LP with linear rates of convergence [24, 32, 47, 70, 71]. On the other hand, the linear rates depend on potentially loose and hard-to-compute constants; hence, tailing off may still be observed in practice. To our knowledge, ours is the first work to combine both theoretical enhancements with practical heuristics, demonstrating their combined effectiveness with extensive computational experiments on standard benchmark instances. In fact, our experiments will expose a substantial gap between algorithms presented in the literature and what's needed to obtain good performance.

Starting from a baseline primal-dual hybrid gradient (PDHG) method [19] applied to a saddle point formulation of LP, we develop a series of algorithmic improvements. These enhancements include adaptive restarting [7], dynamic primal-dual step size selection [36, 37], presolving techniques [1], and diagonal preconditioning (data equilibration) [33]. Most of these enhancements, while inspired by existing literature, are novel. We name our collection of enhancements *PDLP* (PDHG for LP).

The impact of these improvements is substantial. For example, on 383 LP instances derived from the MIPLIB 2017 collection [34], our implementation of a baseline version of PDHG solved only 50 problems to $10^{-8}$ relative accuracy given a limit of approximately 100,000 iterations per problem. By contrast, PDLP solves 283 of the 383 problems under the same conditions. We demonstrate that PDLP outperforms FOM baselines and, in a small number of cases, obtains performance competitive with a commercial LP solver.

Although not the focus of this paper, we believe that our results open the door to a new set of possibilities and computational trade-offs when solving LP problems. PDLP has the potential to solve extremely large scale instances where the simplex method and interior-point methods are unable to run because of their reliance on matrix factorization. Since PDLP uses matrix-vector operations at its core, it can effectively run on multi-threaded CPUs, GPUs [68], or distributed clusters [26]. Furthermore, a GPU implementation of PDLP could efficiently solve batches of similar problems, a setup that has already been successfully applied with other optimization algorithms in applications like strong branching [46] and training neural networks that contain optimization layers [5].

**Outline.** The remainder of this section focuses on related work. Section 2 introduces LP and PDHG. Section 3 describes the set of enhancements that define PDLP. Section 4 presents numerical experiments, and Section 5 concludes and outlines future directions.

## 1.1 Literature review

**PDHG** PDHG was first developed by Zhu and Chan [72], with subsequent analysis and extension by a number of authors [3, 18, 19, 21, 27, 38, 60]. PDHG is closely related to the Arrow-Hurwicz method [8]. PDHG is a form of operator-splitting [11, 64] and can be interpreted as a variant of the alternating directions method of multipliers (ADMM) and Douglas-Rachford splitting (DRS) [16, 25, 56], which themselves are both instantiations of the proximal point method [25, 58, 62]. As opposed to ADMM or DRS, PDHG is 'matrix-free' in that the data matrix is only used for matrix-vector multiplications. This allows PDHG to scale to problems even larger than those tackled by these other techniques, and to make better use of parallel and distributed computation.

**FOM-based solvers** Recent interest in large-scale cone programming has sparked the development several first-order solvers based on competing methods. ProxSDP [66] is a solver for semidefinite programming based on PDHG. Solvers based on Nesterov's accelerated gradients [49] include TFOCS [14], and FOM which is a suite of solvers employing both gradient and proximal algorithms [13]. Solvers based on operator splitting techniques like ADMM include SCS [52–54], OSQP [67], POGS [28], and COSMO [30]. Of these both SCS and POGS offer a matrix-free implementation where the linear system, that arises from the proximal operator used in ADMM, is solved using the conjugate gradient method. However, we shall show experimentally that our method can be significantly faster and more robust than this approach. Finally, [4] considers applying a truncated semismooth Newton method to the system of equations defining a fixed point of the SCS operator.

**FOMs for LP** Lan, Lu and Monteiro [40] and Renegar [61] develop FOMs for LP as a special case of semidefinite programming, with sublinear convergence rates. The FOM-based solvers above all apply to more general problem classes like cone programming or quadratic programming. In contrast, some of the enhancements that constitute PDLP are specialized, either in theory or practice, for LP (namely restarts [7] and presolving). A number of authors [24, 32, 47, 70, 71] have proposed linearly convergent FOMs for LP; to our knowledge, none have been subject of a comprehensive computational study. ECLIPSE [10] solves huge-scale industrial LP problems by accelerated gradient descent, without presenting comparisons on standard test problems. Lin et al. [42] propose an ADMM-based interior point method. In contrast with PDLP which solves to high accuracy (i.e., $10^{-8}$ relative error), [42] perform experiments with $10^{-3}$ and $10^{-5}$ relative error. SNIPAL [41] is a semismooth Newton method based on the proximal augmented Lagrangian. SNIPAL has fast asymptotic convergence, yet, to get good performance, the authors use ADMM for warm-starts. Given PDLP's favorable comparisons with SCS, it's plausible that PDLP could provide a more effective warm-start. Finally, Pock and Chambolle [59] apply PDHG with diagonal preconditioning to a limited set of test LP problems and Applegate et al. [6] show how to extract infeasibility certificates when applying PDHG to LP.

## 2 Preliminaries

In this section, we introduce the notation we use throughout the paper, summarize the LP formulations we solve, and introduce the baseline PDHG algorithm.

**Notation.** Let $\mathbb{R}$ denote the set of real numbers, $\mathbb{R}^+$ the set of nonnegative real numbers, and $\mathbb{R}^-$ the set of nonpositive real numbers. Let $\mathbb{N}$ denote the set of natural numbers (starting from one). Let $\|\cdot\|_p$ denote the $\ell_p$ norm for a vector, and let $\|\cdot\|_2$ denote the spectral norm for a matrix. For a vector $v \in \mathbb{R}^n$, we use $v^+$ and $v^-$ for their positive and negative parts, i.e., $v_i^+ = \max\{0, v_i\}$ and $v_i^- = \min\{0, v_i\}$. The symbol $v_{1:m}$ denotes the vector with the first $m$ components of $v$. The symbols $K_{i,\cdot}$ and $K_{\cdot,j}$ correspond to the $i$th column and $j$th row of the matrix $K$, respectively. The symbol $\mathbf{1}$ denotes the vector of all ones. Given a convex set $X$, we use $\mathbf{proj}_X$ to denote the map that projects onto $X$.

**Linear Programming.** We solve primal-dual LP problems of the form:

$$
\begin{array}{ll}
\begin{aligned}
\underset{x \in \mathbb{R}^n}{\text{minimize}} \quad & c^\top x \\
\text{subject to:} \quad & Gx \geq h \\
& Ax = b \\
& l \leq x \leq u
\end{aligned}
&
\begin{aligned}
\underset{y \in \mathbb{R}^{m_1+m_2}, \lambda \in \mathbb{R}^n}{\text{maximize}} \quad & q^\top y + l^\top \lambda^+ - u^\top \lambda^- \\
\text{subject to:} \quad & c - K^\top y = \lambda \\
& y_{1:m_1} \geq 0 \\
& \lambda \in \Lambda ,
\end{aligned}
\end{array}
\tag{1}
$$

where $G \in \mathbb{R}^{m_1 \times n}, A \in \mathbb{R}^{m_2 \times n}, c \in \mathbb{R}^n, h \in \mathbb{R}^{m_1}, b \in \mathbb{R}^{m_2}, l \in (\mathbb{R} \cup \{-\infty\})^n, u \in (\mathbb{R} \cup \{\infty\})^n$, $K^\top = (G^\top, A^\top), q^\top := (h^\top, b^\top)$, and

$$
\Lambda = \Lambda_1 \times \cdots \times \Lambda_n \quad \Lambda_i := \begin{cases} \{0\} & l_i = -\infty, \ u_i = \infty, \\ \mathbb{R}^- & l_i = -\infty, \ u_i \in \mathbb{R} \\ \mathbb{R}^+ & l_i \in \mathbb{R}, \ u_i = \infty \\ \mathbb{R} & \text{otherwise} \end{cases}
$$

is the set of variables $\lambda$ such that the dual objective is finite. This pair of primal-dual problems is equivalent to the saddle-point problem:

$$
\min_{x \in X} \max_{y \in Y} \mathcal{L}(x, y) := c^\top x - y^\top K x + q^\top y
\tag{2}
$$

with $X := \{x \in \mathbb{R}^n : l \leq x \leq u\}$, and $Y := \{y \in \mathbb{R}^{m_1+m_2} : y_{1:m_1} \geq 0\}$.

**PDHG.** When specialized to (2), the PDHG algorithm takes the form:

$$
\begin{aligned}
x^{k+1} &= \mathbf{proj}_X(x^k - \tau(c - K^\top y^k)) \\
y^{k+1} &= \mathbf{proj}_Y(y^k + \sigma(q - K(2x^{k+1} - x^k)))
\end{aligned}
\tag{3}
$$

where $\tau, \sigma > 0$ are primal and dual step sizes, respectively. PDHG is known to converge to an optimal solution when $\tau\sigma\|K\|_2^2 \leq 1$ [20, 21]. We reparameterize the step sizes by

$$\tau = \eta/\omega \quad \text{and} \quad \sigma = \omega\eta \qquad \text{with } \eta \in (0, \infty) \quad \text{and} \quad \omega \in (0, \infty). \tag{4}$$

We call $\omega \in (0, \infty)$ the *primal weight*, and $\eta \in (0, \infty)$ the *step size*. Under this reparameterization PDHG converges for all $\eta \leq 1/\|K\|_2$. This allows us to control the scaling between the primal and dual iterates with a single parameter $\omega$. We use the term *primal weight* to describe $\omega$ because it weights the primal variables in the following norm:

$$\|z\|_\omega := \sqrt{\omega\|x\|_2^2 + \frac{\|y\|_2^2}{\omega}}.$$

This norm plays a role in the theory for PDHG [20] and later algorithmic discussions.

For the *baseline PDHG algorithm* that we use for comparisons, we consider two simple choices for $\eta$ and $\omega$. For the step size, set $\eta = 0.9/\|K\|_2$ where $\|K\|_2$ is estimated via power iteration, and for the primal weight we set $\omega = 1$; this is similar to the default parameters in the standard PDHG implementation in ODL [2].

## 3    Practical algorithmic improvements

In this section, we detail these enhancements, and defer further experimental testing of them to Section 4 and ablation studies to Appendix C. While our enhancements are inspired by theory, our focus is on practical performance. The algorithm as a whole has no convergence guarantee, although some individual enhancements do; see Section 3.6 for further discussion.

Algorithm 1 presents pseudo-code for PDLP after preprocessing steps. We modify the step sizes (Section 3.1), add restarts (Section 3.2), and dynamically update the primal weights (Section 3.3). Before running Algorithm 1 we apply presolve (Section 3.4) and diagonal preconditioning (Section 3.5). There are some minor differences between the pseudo-code and the actual code. In particular, we only evaluate the restart or termination criteria (Line 10) every $40$ iterations. This reduces the associated overheads with minimal impact on the total number of iterations. We also check the termination criteria before beginning the algorithm or if we detect a numerical error.

---

**Algorithm 1:** PDLP (after preconditioning and presolve)

1   **Input:** An initial solution $z^{0,0}$;

2   Initialize outer loop counter $n \leftarrow 0$, total iterations $k \leftarrow 0$, step size $\hat{\eta}^{0,0} \leftarrow 1/\|K\|_\infty$, primal weight $\omega^0 \leftarrow \texttt{InitializePrimalWeight}(c, q)$;

3   **repeat**

4      $t \leftarrow 0$;

5      **repeat**

6          $z^{n,t+1}, \eta^{n,t+1}, \hat{\eta}^{n,t+1} \leftarrow \texttt{AdaptiveStepOfPDHG}(z^{n,t}, \omega^n, \hat{\eta}^{n,t}, k)$ ;

7          $\bar{z}^{n,t+1} \leftarrow \frac{1}{\sum_{i=1}^{t+1} \eta^{n,i}} \sum_{i=1}^{t+1} \eta^{n,i} z^{n,i}$ ;

8          $z_{\mathrm{c}}^{n,t+1} \leftarrow \texttt{GetRestartCandidate}(z^{n,t+1}, \bar{z}^{n,t+1}, z^{n,0})$ ;

9          $t \leftarrow t + 1, k \leftarrow k + 1$ ;

10      **until** *restart or termination criteria holds*;

11      **restart the outer loop.** $z^{n+1,0} \leftarrow z_{\mathrm{c}}^{n,t}, n \leftarrow n + 1$;

12      $\omega^n \leftarrow \texttt{PrimalWeightUpdate}(z^{n,0}, z^{n-1,0}, \omega^{n-1})$ ;

13   **until** *termination criteria holds*;

14   **Output:** $z^{n,0}$.

---

### 3.1    Step size choice

The convergence analysis [20, Equation (15)] of PDHG (equation (3)) relies on a small constant step size

$$\eta \leq \frac{\|z^{k+1} - z^k\|_\omega^2}{2(y^{k+1} - y^k)^\top K(x^{k+1} - x^k)} \tag{5}$$

**Algorithm 2:** One step of PDHG using our step size heuristic

**1** **Function** AdaptiveStepOfPDHG ($z^{n,t}$, $\omega^n$, $\hat{\eta}^{n,t}$, $k$)**:**

**2** $\quad$ $(x, y) \leftarrow z^{n,t}, \eta \leftarrow \hat{\eta}^{n,t}$ ;

**3** $\quad$ **for** $i = 1, \ldots, \infty$ **do**

**4** $\quad\quad$ $x' \leftarrow \mathbf{proj}_X(x - \frac{\eta}{\omega^n}(c - K^\top y))$ ;

**5** $\quad\quad$ $y' \leftarrow \mathbf{proj}_Y(y + \eta\omega^n(q - K(2x' - x)))$ ;

**6** $\quad\quad$ $\bar{\eta} \leftarrow \frac{\|(x'-x, y'-y)\|^2_{\omega^n}}{2(y'-y)^\top K(x'-x)}$ ;

**7** $\quad\quad$ $\eta' \leftarrow \min\left((1 - (k+1)^{-0.3})\bar{\eta}, (1 + (k+1)^{-0.6})\eta\right)$ ;

**8** $\quad\quad$ **if** $\eta \leq \bar{\eta}$ **then**

**9** $\quad\quad\quad$ **return** $(x', y')$, $\eta$, $\eta'$

**10** $\quad\quad$ **end**

**11** $\quad\quad$ $\eta \leftarrow \eta'$ ;

**12** $\quad$ **end**

where $z^k = (x^k, y^k)$. Classically one would ensure (5) by picking $\eta = \frac{1}{\|K\|_2}$. This is overly pessimistic and requires estimation of $\|K\|_2$. Instead our AdaptiveStepOfPDHG adjusts $\eta$ dynamically to ensure that (5) is satisfied. If (5) isn't satisfied, we abort the step; i.e., we reduce $\eta$, and try again. If (5) is satisfied we accept the step. This is described in Algorithm 2. Note that in Algorithm 2 $\bar{\eta} \geq \frac{1}{\|K\|_2}$ holds always, and from this one can show the resulting step size $\eta \geq \frac{1-o(1)}{\|K\|_2}$ holds as $k \to \infty$.

Our step size routine compares favorably in practice with the line search by Malitsky and Pock [43] (See Appendix C.1).

### 3.2 Adaptive restarts

In PDLP, we adaptively restart the PDHG algorithm in each outer iteration. The key to our restarts at the $n$-th outer iteration is the normalized duality gap at $z$ which for any radius $r \in (0, \infty)$ is defined by

$$\rho_r^n(z) := \frac{1}{r} \underset{(\hat{x}, \hat{y}) \in \{\hat{z} \in Z : \|\hat{z} - z\|_{\omega^n} \leq r\}}{\text{maximize}} \{\mathcal{L}(x, \hat{y}) - \mathcal{L}(\hat{x}, y)\},$$

introduced by [7]. Unlike the standard duality gap

$$\underset{(\hat{x}, \hat{y}) \in Z}{\text{maximize}} \{\mathcal{L}(x, \hat{y}) - \mathcal{L}(\hat{x}, y)\},$$

the normalized duality gap is always a finite quantity. Furthermore, for any value of $r$ and $\omega^n$, the normalized duality gap $\rho_r^n(z)$ is 0 if and only if the solution $z$ is an optimal solution to (2) [7]; thus, it provides a valid metric for measuring progress towards the optimal solution. The normalized duality gap is computable in linear time [7]. For brevity, define $\mu_n(z, z_{\text{ref}})$ as the normalized duality gap at $z$ with radius $\|z - z_{\text{ref}}\|_{\omega^n}$, i.e.,

$$\mu_n(z, z_{\text{ref}}) := \rho_{\|z-z_{\text{ref}}\|_{\omega^n}}^n(z),$$

where $z_{\text{ref}}$ is a user-chosen reference point.

**Choosing the restart candidate.** To choose the restart candidate $z_{\text{c}}^{n,t+1}$ we call

$$\texttt{GetRestartCandidate}(z^{n,t+1}, \bar{z}^{n,t+1}, z^{n,0}) := \begin{cases} z^{n,t+1} & \mu_n(z^{n,t+1}, z^{n,0}) < \mu_n(\bar{z}^{n,t+1}, z^{n,0}) \\ \bar{z}^{n,t+1} & \text{otherwise} \end{cases}.$$

This choice is justified in Remark 5 of [7].

**Restart criteria.** We define three parameters: $\beta_{\text{sufficient}} \in (0, 1)$, $\beta_{\text{necessary}} \in (0, \beta_{\text{sufficient}})$ and $\beta_{\text{artificial}} \in (0, 1)$. In PDLP we use $\beta_{\text{sufficient}} = 0.9$, $\beta_{\text{necessary}} = 0.1$, and $\beta_{\text{artificial}} = 0.5$. The algorithm restarts if one of three conditions holds:

(i) (**Sufficient decay in normalized duality gap**) $\mu_n(z_{\text{c}}^{n,t+1}, z^{n,0}) \leq \beta_{\text{sufficient}}\mu_n(z^{n,0}, z^{n-1,0})$ ,

(ii) (**Necessary decay + no local progress in normalized duality gap**)

$\quad$ $\mu_n(z_{\text{c}}^{n,t+1}, z^{n,0}) \leq \beta_{\text{necessary}}\mu_n(z^{n,0}, z^{n-1,0})$ $\quad$ and $\quad$ $\mu_n(z_{\text{c}}^{n,t+1}, z^{n,0}) > \mu_n(z_c^{n,t}, z^{n,0})$ ,

(iii) (**Long inner loop**) $t \geq \beta_{\text{artificial}} k$ .

The motivation for (i) is presented in [7]; it guarantees the linear convergence of restarted PDHG on LP problems. The second condition in (ii) is inspired by adaptive restart schemes for accelerated gradient descent where restarts are triggered if the function value increases [57]. The first inequality in (ii) provides a safeguard for the second one, preventing the algorithm restarting every inner iteration or never restarting. The motivation for (iii) relates to the primal weights (Section 3.3). In particular, primal weight updates only occur after a restart, and condition (iii) ensures that the primal weight will be updated infinitely often. This prevents a bad choice of primal weight in earlier iterations causing progress to stall for a long time.

### 3.3 Primal weight updates

The primal weight is initialized using

$$\texttt{InitializePrimalWeight}(c, q) := \begin{cases} \frac{\|c\|_2}{\|q\|_2} & \|c\|_2, \|q\|_2 > \epsilon_{\text{zero}} \\ 1 & \text{otherwise} \end{cases}$$

where $\epsilon_{\text{zero}}$ is a small nonzero tolerance. This primal weight update scheme guarantees scale invariance. In particular, in Appendix A we consider PDHG with $\epsilon_{\text{zero}} = 0$, $\eta = 0.9/\|K\|_2$ and $\omega = \texttt{InitializePrimalWeight}(c, q)$. In this simplified setting, we prove that if we multiply the objective, constraints, or the right hand side and variable bounds by a scalar then the iterate behaviour remain identical (up to a scaling factor).

---

**Algorithm 3:** Primal weight update

---

1 **Function** `PrimalWeightUpdate`($z^{n,0}, z^{n-1,0}, \omega^{n-1}$)**:**

2     $\Delta_x^n = \|x^{n,0} - x^{n-1,0}\|_2, \quad \Delta_y^n = \|y^{n,0} - y^{n-1,0}\|_2$ ;

3     **if** $\Delta_x^n > \epsilon_{zero}$ *and* $\Delta_y^n > \epsilon_{zero}$ **then**

4        **return** $\exp\left( \theta \log\left( \frac{\Delta_y^n}{\Delta_x^n} \right) + (1 - \theta) \log\left( \omega^{n-1} \right) \right)$

5     **else**

6        **return** $\omega^{n-1}$ ;

7     **end**

---

Algorithm 3 aims to choose the primal weight $\omega^n$ such that distance to optimality in the primal and dual is the same, i.e., $\|(x^{n,t} - x^\star, \mathbf{0})\|_{\omega^n} \approx \|(\mathbf{0}, y^{n,t} - y^\star)\|_{\omega^n}$. By definition of $\| \cdot \|_\omega$,

$$\|(x^{n,t} - x^\star, \mathbf{0})\|_{\omega^n} = \omega^n \|x^{n,t} - x^\star\|_2, \quad \|(\mathbf{0}, y^{n,t} - y^\star)\|_{\omega^n} = \frac{1}{\omega^n} \|y^{n,t} - y^\star\|_2.$$

Setting these two terms equal yields $\omega^n = \frac{\|y^{n,t} - y^\star\|_2}{\|x^{n,t} - x^\star\|_2}$. Of course, the quantity $\frac{\|y^{n,t} - y^\star\|_2}{\|x^{n,t} - x^\star\|_2}$ is unknown beforehand, but we attempt to estimate it using $\Delta_y^n / \Delta_x^n$. However, the quantity $\Delta_y^n / \Delta_x^n$ can change wildly from one restart to another, causing $\omega^n$ to oscillate. To dampen variations in $\omega^n$, we first move to a log-scale where the primal weight is symmetric, i.e., $\log(1/\omega^n) = -\log(\omega^n)$, and perform a exponential smoothing with parameter $\theta \in [0, 1]$. In PDLP, we use $\theta = 0.5$.

There are several important differences between our primal weight heuristic and literature [36, 37]. For example, [36, 37] make relatively small changes to the primal weights at each iteration, attempting to balance the primal and dual residual. These changes have to be diminishingly small because, in our experience, PDHG may be unstable if they are too big. In contrast, in our method the primal weight is only updated during restarts, which in practice allows for much larger changes without instability issues. Moreover, our scheme tries to balance the weighted distance traveled in the primal and dual rather than the residuals [36, 37].

### 3.4 Presolve

Presolving refers to transformation steps that simplify the input problem before starting the optimization solver. These steps span from relatively easy transformations such as detecting inconsistent bounds, removing empty rows and columns of $K$, and removing variables whose lower and upper bounds are equal, to more complex operations such as detecting duplicate rows in $K$ and tightening bounds. Presolve is a standard component of traditional LP solvers [44]. We are not aware of presolve being combined with PDHG for LP. However, [41, 42] combine presolve with other FOMs.

As an experiment to measure the impact of presolve, we used PaPILO [29], an open-source presolving library. For technical reasons, it was easier to use PaPILO as a standalone executable than as a library. We simulate its effect by simply solving the preprocessed instances. Convergence criteria are evaluated with respect to the presolved instance, not the original problem.

## 3.5 Diagonal Preconditioning

Preconditioning is a popular heuristic in optimization for improving the convergence of FOMs. To avoid factorizations, we only consider diagonal preconditioners. Our goal is to rescale the constraint matrix $K = (G, A)$ to $\tilde{K} = (\tilde{G}, \tilde{A}) = D_1 K D_2$ with positive diagonal matrices $D_1$ and $D_2$, so that the resulting matrix $\tilde{K}$ is "well balanced". Such preconditioning creates a new LP instance that replaces $A, G, c, b, h, u$, and $l$ in (1) with $\tilde{G}, \tilde{A}, \hat{x} = D_2^{-1}x, \tilde{c} = D_2 c, (\tilde{b}, \tilde{h}) = D_1(b, h), \tilde{u} = D_2^{-1}u$ and $\tilde{l} = D_2^{-1}l$. Common choices for $D_1$ and $D_2$ include:

- **No scaling:** Solve the original LP instance (1) without additional scaling, namely $D_1 = D_2 = I$.

- **Pock-Chambolle [59]:** Pock and Chambolle proposed a family of diagonal preconditioners[3] for PDHG parameterized by $\alpha$, where the diagonal matrices are defined by $(D_1)_{jj} = \sqrt{\|K_{j,\cdot}\|_{2-\alpha}}$ for $j = 1, ..., m_1 + m_2$ and $(D_2)_{ii} = \sqrt{\|K_{\cdot,i}\|_{\alpha}}$ for $i = 1, ..., n$. We use $\alpha = 1$ in PDLP (we also tested $\alpha = 0$ and $\alpha = 2$). This is the baseline diagonal preconditioner in the PDHG literature.

- **Ruiz [63]:** Ruiz scaling is a popular algorithm in numerical linear algebra to equilibrate matrices. In an iteration of Ruiz scaling, the diagonal matrices are defined as $(D_1)_{jj} = \sqrt{\|K_{j,\cdot}\|_{\infty}}$ for $j = 1, ..., m_1 + m_2$ and $(D_2)_{ii} = \sqrt{\|K_{\cdot,i}\|_{\infty}}$ for $i = 1, ..., n$. Ruiz [63] shows that if this rescaling is applied iteratively, the infinity norm of each row and each column converge to 1.

For the default PDLP settings, we apply a combination of Ruiz rescaling [63] and the preconditioning technique proposed by Pock and Chambolle [59]. In particular, we apply 10 iterations of Ruiz scaling and then apply the Pock-Chambolle scaling. To illustrate the effectiveness of our proposed scaling technique, we compare it against these three common techniques in Appendix C.5.

## 3.6 Theoretical guarantees for the above enhancements

While PDLP's enhancements are motivated by theory, some of them may not preserve theoretical guarantees as discussed below:

- We do not have a proof of convergence for the adaptive step size rule (Section 3.1).

- One can show our restart criteria (Section 3.2) preserve convergence guarantees by modifying the proof of [7] to a more general setting.

- Primal weight updates (Section 3.3) do not readily preserve convergence guarantees, but we conjecture that a proof of convergence is possible if they are updated infrequently.

- Presolve (Section 3.4) and diagonal preconditioning (Section 3.5) preserve theoretical guarantees because they can be viewed as applying PDHG to an LP instance with different data.

## 4 Numerical experiments

Our numerical experiments study the effectiveness of PDLP primarily with respect to traditional LP applications and benchmark sets. Section 4.1 describes the setup for the experiments. Section 4.2 demonstrates PDLP's improvements over baseline PDHG. Section 4.3 compares PDLP with other FOMs. Section 4.4 highlights benchmark instances where PDLP outperforms a commercial LP solver. Finally, Section 4.5 illustrates the ability of PDLP to scale to a large application where barrier and simplex-based solvers run out of memory. The supplemental materials contain extensive ablation studies and additional instructions for reproducing the experiments.

---

[3]Diagonal preconditioning is equivalent to changing to a weighted $\ell_2$ norm in the proximal step of PDHG (weight defined by $D_2$ and $D_1$ for the primal and dual respectively). Pock and Chambolle use this weighted norm perspective.

## 4.1 Experimental setup

**Optimality termination criteria.** PDLP terminates with an approximately optimal solution when the primal-dual iterates $x \in X, y \in Y, \lambda \in \Lambda$, satisfy:

$$\left| q^\top y + l^\top \lambda^+ - u^\top \lambda^- - c^\top x \right| \leq \epsilon (1 + \left| q^\top y + l^\top \lambda^+ - u^\top \lambda^- \right| + \left| c^\top x \right|) \tag{6a}$$

$$\left\| \begin{pmatrix} Ax - b \\ (h - Gx)^+ \end{pmatrix} \right\|_2 \leq \epsilon (1 + \|q\|_2) \tag{6b}$$

$$\|c - K^\top y - \lambda\|_2 \leq \epsilon (1 + \|c\|_2) \tag{6c}$$

where $\epsilon \in (0, \infty)$ is the termination tolerance. Note that if (6) is satisfied with $\epsilon = 0$, then by LP duality we have found an optimal solution [35]. Indeed, (6a) is the duality gap, (6b) is primal feasibility, and (6c) is dual feasibility. We use these criteria to be consistent with those of SCS [54]. The PDHG algorithm does not explicitly include a reduced costs variable $\lambda$. Therefore, to evaluate the optimality termination criteria we compute $\lambda = \mathbf{proj}_\Lambda(c - K^\top y)$. All instances considered have an optimal primal-dual solution. We use $\epsilon = 10^{-8}$ as a benchmark for high-quality solutions and $\epsilon = 10^{-4}$ for moderately accurate solutions.

**Benchmark datasets.** We use three datasets to compare algorithmic performance. One is the `LP benchmark` dataset of 56 problems, formed by merging the instances from "Benchmark of Simplex LP Solvers", "Benchmark of Barrier LP solvers", and "Large Network-LP Benchmark" from [45]. We also created a larger benchmark of 383 instances curated from LP relaxations of mixed-integer programming problems from the MIPLIB2017 collection [34] (see Appendix B) that we label `MIP Relaxations`. `MIP Relaxations` was used extensively during algorithmic development, e.g., for hyperparameter choices; we held out `LP benchmark` as a test set. Finally, we also performed some experiments on the `Netlib` LP benchmark [31], an historically important benchmark that is no longer state of the art for large-scale LP.

**Software.** PDLP is implemented in an open-source Julia [15] module available at `https://github.com/google-research/FirstOrderLp.jl`. The module also contains a baseline implementation of the extragradient method with many of the same enhancements as PDLP (labeled 'Enh. Extragradient'). We compare with two external packages: SCS [54] version 2.1.3, an open-source generic cone solver based on ADMM, and Gurobi version 9.0.1, a state-of-the-art commercial LP solver. SCS supports two modes for solving the linear system that arises at each iteration, a direct method based on a cached LDL factorization (which is the default 'SCS') and an indirect method based on the conjugate gradient method (which we label 'SCS (matrix-free)'). All solvers are run single-threaded. SCS and Gurobi are provided the same presolved instances as PDLP.

**Computing environment.** We used two computing environments for our experiments: 1) `e2-highmem-2` virtual machines (VMs) on Google Cloud Platform (GCP). Each VM provides two virtual CPUs and 16GB RAM. 2) A dedicated workstation with an Intel Xeon E5-2669 v3 processor and 128 GB RAM. This workstation has a license for Gurobi that permits at most one concurrent solve. Total compute time on GCP for all preliminary and final experiments was approximately $72,000$ virtual CPU hours.

**Initialization.** All first-order methods use all-zero vectors as the initial starting points.

**Metrics.** We use the term *KKT passes* to refer to the number of matrix multiplications by both $K$ and $K^\top$. Given that the most expensive operation in our algorithm is matrix-vector multiplication, this metric is less noisy than runtime for comparing performance between matrix-free solvers. SGM10 stands for shifted geometric mean with shift 10, which is computed by adding 10 to all data points, taking the geometric mean, and then subtracting 10. Unsolved instances are assigned values corresponding to the limits specified in the next paragraph.

**Time and KKT pass limits.** For Section 4.2 we impose a limit on the KKT passes of $100,000$. For Section 4.3 we impose a time limit of 1 hour.

## 4.2 Impact of PDLP's improvements

The y-axes of Figure 1 display the SGM10 of the KKT passes normalized by the value for baseline PDHG. We can see, with the exception of presolve for `LP benchmark` at tolerance $10^{-4}$, each of our modifications described in Section 3 improves the performance of PDHG.

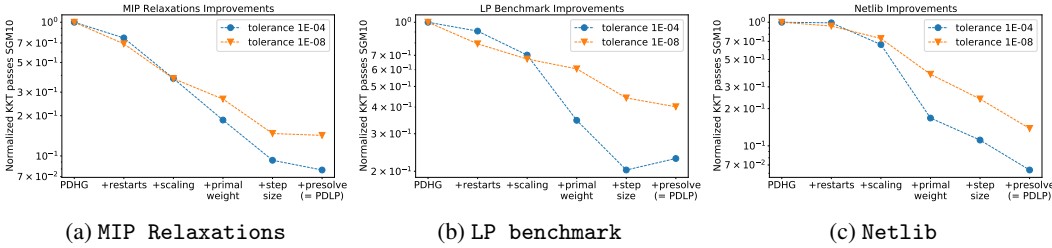

(a) MIP Relaxations    (b) LP benchmark    (c) Netlib

Figure 1: Summary of relative impact of PDLP's improvements

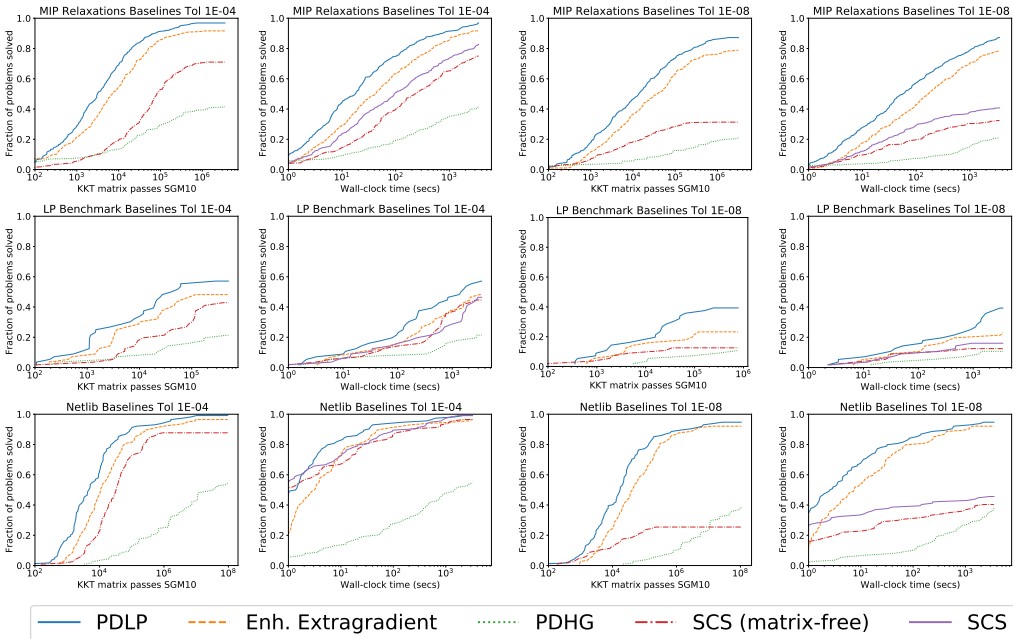

Figure 2: Number of problems solved for MIP Relaxations (top), LP benchmark (middle), and Netlib (bottom) datasets.

## 4.3 Comparison with other first-order baselines

We compared PDLP with several other first-order baselines: SCS [54], in both direct (default) mode and matrix-free mode, and our enhanced implementation of the extragradient method [39, 48]. For SCS in matrix-free mode, we include the KKT passes from the conjugate gradient solves; for SCS in direct mode there is no reasonable measure of KKT passes for the factorization and direct solve, so we only measure running time. The comparisons are summarized in Figure 2.

## 4.4 PDLP versus simplex and barrier

In this section, we test the performance of PDLP against the three methods available in Gurobi: barrier, primal simplex, and dual simplex. By default when provided multiple threads, Gurobi runs these three methods concurrently and terminates when the first method completes. We used default termination for Gurobi and set $\epsilon = 10^{-8}$ for PDLP. We ran experiments with instances from the MIP Relaxations and LP benchmark. Although, for most instances, Gurobi outperforms PDLP, we found problems for which PDLP exhibits moderate to significant gains. Table 1 gives examples of instances where our prototype implementation is within a factor of two of the best of the three Gurobi methods. While further improvements are needed for PDLP to truly compete with the portfolio of methods that Gurobi offers, we interpret these results as evidence that PDLP itself could be of value in this portfolio.

Table 1: Instances from `MIP Relaxations` (top) and `LP benchmark` (bottom) where PDLP is within a factor of 2 of the best of all Gurobi methods. Time to solve in seconds.

| Instance | PDLP | Gurobi Barrier | Gurobi Primal Simp. | Gurobi Dual Simp. |
|---|---|---|---|---|
| ex9 | 1.6 | 102.6 | 181.3 | 47.6 |
| genus-sym-g62-2 | 2.1 | 10.7 | 6.7 | 33.2 |
| highschool1-aigio | 72.6 | 243.8 | >3600 | >3600 |
| neos-578379 | 1.4 | 0.7 | 1.7 | 1.8 |
| rwth-timetable | 1870.3 | >3600 | >3600 | >3600 |
| ex10 | 4.9 | 63.1 | 16.8 | 7.9 |
| nug08-3rd | 2.2 | 3.2 | 2219.2 | 24.1 |
| savsched1 | 35.9 | 25.9 | 56.0 | 261.3 |

Table 2: Solve time for PageRank instances. Gurobi barrier has crossover disabled, 1 thread. PDLP and SCS solve to $10^{-8}$ relative accuracy. SCS is matrix-free. Baseline PDHG is unable to solve any instances. Presolve not applied. OOM = Out of Memory. The number of nonzero coefficients per instance is $8 \times (\# \text{ nodes}) - 18$.

| # nodes | PDLP | SCS | Gurobi Barrier | Gurobi Primal Simp. | Gurobi Dual Simp. |
|---|---|---|---|---|---|
| $10^4$ | 7.4 sec. | 1.3 sec. | 36 sec. | 37 sec. | 114 sec. |
| $10^5$ | 35 sec. | 38 sec. | 7.8 hr. | 9.3 hr. | >24 hr. |
| $10^6$ | 11 min. | 25 min. | OOM | >24 hr. | - |
| $10^7$ | 5.4 hr. | 3.8 hr. | - | - | - |

### 4.5 Large-scale application: PageRank

Nesterov [50, equation (7.3)] gives an LP formulation of the standard "PageRank" problem. Although the LP formulation is not the best approach to computing PageRank, it is a source of very large instances. For a random scalable collection of PageRank instances, we used Barabási-Albert [9] preferential attachment graphs with approximately three edges per node; see Appendix D for details. The results are summarized in Table 2.

## 5 Conclusions and future work

We find our experimental results encouraging for the application of FOMs like PDHG to LP. At a minimum, they provide evidence against the claim that FOMs are useful only when moderately accurate solutions are desired. The practical success of our heuristics that lack theoretical guarantees provides fresh motivation for theoreticians to study these methods. It is important, as well, to understand what drives the difficulty of some instances and how they could be transformed to solve more quickly. We hope the community will use the benchmarks and baselines released with this work as a starting point for further investigating new FOMs for LP. With additional algorithmic and implementation refinements, we believe that PDLP or similar approaches could become part of the standard toolkit for linear programming.

## Acknowledgments and Disclosure of Funding

We thank Yura Malitsky for advice on parameter choices for the linesearch rule of [43].

The authors have no third-party funding or competing interests to declare.

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
