# A Proof of scale invariance of primal weight initialization scheme

**Proposition 1.** *Suppose that $\hat{K} = \gamma K$, $\hat{c} = \gamma \alpha_y c$, $\hat{q} = \gamma \alpha_x q$, $\hat{l} = \alpha_x l$, and $\hat{u} = \alpha_x u$ for $\alpha_y, \alpha_x, \gamma \in (0, \infty)$ with $\|c\|_2, \|q\|_2, \|\hat{c}\|_2, \|\hat{q}\|_2 > \epsilon_{mach}$. Consider the PDHG algorithm given in (3) with $\omega = \texttt{InitializePrimalWeight}(c, q)$. Let $z^k$ be the PDHG iterates on the original problem and $\hat{z}^k$ be the PDHG iterates on the scaled problem with $\hat{x}^0 = \alpha_x x^0$ and $\hat{y}^0 = \alpha_y y^0$, then: $\hat{x}^k = \alpha_x x^k, \hat{y}^k = \alpha_y y^k$ for all $k \in \{0\} \cup \mathbb{N}$.*

*Proof.* We will prove this by induction. By definition the result holds for $k = 0$. Define $\hat{\eta} = \eta/\gamma$, and $\hat{\omega} = \|\hat{c}\|/\|\hat{q}\|_2 = \omega(\alpha_y/\alpha_x)$. Then,

$$
\begin{aligned}
\hat{x}^{k+1} &= \mathop{\textbf{proj}}_{\hat{X}} \left( \hat{x}^k - \hat{\eta}/\hat{\omega}(\hat{c} - \hat{K}^\top \hat{y}^k) \right) \\
&= \mathop{\textbf{proj}}_{\hat{X}} \left( \alpha_x x^k - \alpha_x \eta/\omega(c - K^\top y^k) \right) \\
&= \alpha_x \mathop{\textbf{proj}}_{X} \left( x^k - \eta/\omega(c - K^\top y^k) \right) \\
&= \alpha_x x^{k+1}.
\end{aligned}
$$

Similarly,

$$
\begin{aligned}
\hat{y}^{k+1} &= \mathop{\textbf{proj}}_{\hat{Y}} \left( \hat{y}^k - \hat{\eta}\hat{\omega}(\hat{q} - \hat{K}(2\hat{x}^{k+1} - \hat{x}^k)) \right) \\
&= \mathop{\textbf{proj}}_{\hat{Y}} \left( \alpha_y y^k - \alpha_y \eta\omega(q - K(2x^{k+1} - x^k)) \right) \\
&= \alpha_y \mathop{\textbf{proj}}_{Y} \left( \alpha_y y^k - \alpha_y \eta\omega(q - K(2x^{k+1} - x^k)) \right) \\
&= \alpha_y y^{k+1}.
\end{aligned}
$$

$\square$

# B `MIP Relaxations` dataset

MIPLIB 2017 [34] is a collection of mixed integer programming (MIP) problems used primarily for developing and benchmarking MIP solvers. MIPLIB contains both a larger collection set (1056 instances) and a smaller benchmark set (240 instances). We select 383 instances from the collection set that satisfy the following criteria:

- Not tagged as numerically unstable
- Not tagged as infeasible
- Not tagged as having indicator constraints
- Finite optimal objective (if known)
- The constraint matrix has between $100,000$ and $10,000,000$ nonzero coefficients.

For comparison, the MIPLIB benchmark set excludes instances whose constraint matrix has more than $1,000,000$ nonzero coefficients. The upper limit of $10,000,000$ was chosen for the convenience of running experiments. Our set both excludes small instances that may be in the benchmark set and includes instances deemed too large for the benchmark set. From each MIP instance we derive an LP instance by removing the integrality constraints.

# C Ablation study

To study the impact of PDLP's improvements over baseline PDHG, we performed an ablation study, in which we evaluate the consequences of disabling each enhancement separately and evaluate alternative choices. All experiments in this section are performed on the `MIP Relaxations` dataset. Each of these experiments is run with a limit of 100,000 KKT passes and 6 hours. If the instance is unsolved, the KKT passes are set to 100,000, and the solve time to 6 hours.

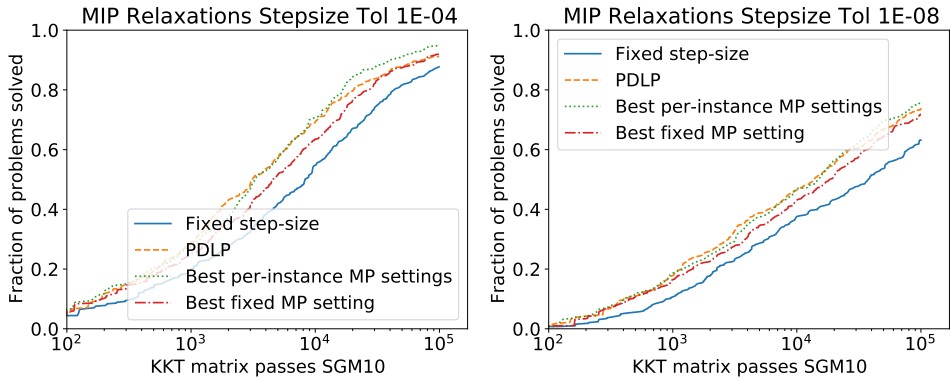

Figure 3: Step size ablation experiments on `MIP Relaxations`

Table 3: Performance statistics: MIP Relaxations Stepsize Tol 1E-04

| Experiment | Solved count | KKT passes SGM10 | Solve time secs SGM10 |
|---|---|---|---|
| Fixed step-size | 336 | 6207.9 | 91.3 |
| PDLP | 349 | 3058.3 | 51.1 |
| Best fixed MP setting | 352 | 3855.6 | 58.5 |
| Best per-instance MP settings | 363 | 2869.7 | 41.4 |

### C.1 Step size choice

We compare PDLP's adaptive step size rule against three alternatives:

- "Fixed step size": (baseline PDHG) The step size $\eta$ is fixed to $\eta = 0.9/\|K\|_2$ where $\|K\|_2$ is estimated via power iteration,

- "Best fixed Malitsky-Pock (MP) setting": Malitsky and Pock [43], tuning the hyperparameters via a hyperparameter search, and

- "Best per-instance Malitsky-Pock (MP) setting": Malitsky and Pock [43], choosing the best hyperparameters separately for each instance. This is a "virtual" solver that combines 42 hyperparameter configurations.

The results, in Figure 3 and Tables 3 and 4, show that PDLP is slightly better than tuned Malitsky-Pock, and at high accuracy, almost as good as per-instance tuned Malitsky-pock.

**Description of Malitsky and Pock hyperparameters.** Our implementation depends on three hyperparameters: `breaking_factor`, `downscaling_factor`, and `interpolation_coefficient`. We explain the role of each one by summarizing the linesearch rule. Suppose the algorithm finished iteration $k$ and it does not execute a restart. Thus, the primal weight doesn't not change $\omega_k = \omega_{k+1}$. Mimicking the notation in [43] we define:

$$\theta_k = \frac{\eta_{k-1}}{\eta_k}.$$

Then, the algorithm does the following at iteration $k + 1$:

1. Update primal iterate $x^{k+1} \leftarrow \mathbf{proj}_X \left( x^k - \frac{\eta_k}{\omega_k} \left( c - K^\top y^k \right) \right).$

Table 4: Performance statistics: MIP Relaxations Stepsize Tol 1E-08

| Experiment | Solved count | KKT passes SGM10 | Solve time secs SGM10 |
|---|---|---|---|
| Fixed step-size | 242 | 17339.5 | 469.9 |
| Best fixed MP setting | 275 | 11660.4 | 260.1 |
| PDLP | 283 | 9773.3 | 216.0 |
| Best per-instance MP settings | 289 | 9778.7 | 193.8 |

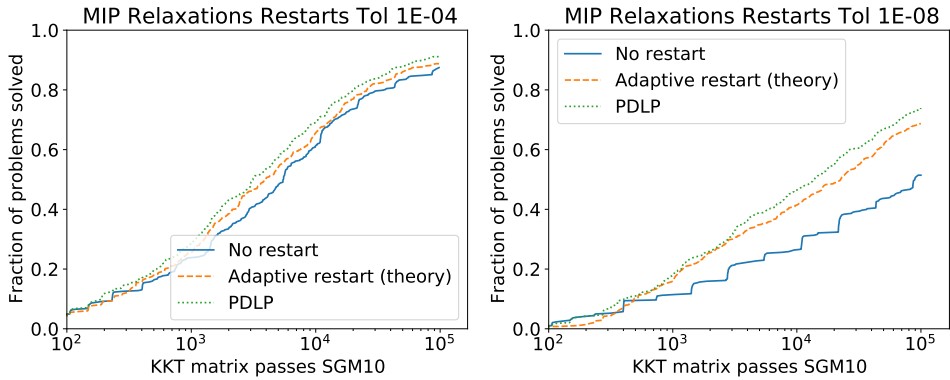

Figure 4: Restart ablation experiments on `MIP Relaxations`

Table 5: Performance statistics: MIP Relaxations Restarts Tol 1E-04

| Experiment | Solved count | KKT passes SGM10 | Solve time secs SGM10 |
|---|---|---|---|
| No restart | 335 | 4387.9 | 70.2 |
| Adaptive restart (theory) | 340 | 3680.5 | 64.0 |
| PDLP | 349 | 3058.3 | 51.1 |

2. Pick a candidate for the next step size $\widehat{\eta}_{k+1} \in [\eta_k, \sqrt{1+\theta_k}\eta_k]$. By letting

$$\widehat{\eta}_{k+1} \leftarrow \eta_k + \texttt{interpolation\_coefficient} \cdot \left(\sqrt{1+\theta_k} - 1\right)\eta_k \quad \text{and} \quad \widehat{\theta}_{k+1} \leftarrow \frac{\eta_k}{\widehat{\eta}_{k+1}}.$$

3. Compute a candidate for next dual iterate $y_{k+1}$:

$$\widehat{y}^{k+1} \leftarrow \underset{Y}{\textbf{proj}} \left(y^k + \omega_{k+1}\widehat{\eta}_{k+1}\left(q - K\left(x^{k+1} + \widehat{\theta}_{k+1}(x^{k+1} - x^k)\right)\right)\right).$$

4. Check if the linesearch is done;

   **If** $\widehat{\eta}_k \|K^\top(\widehat{y}^{k+1} - y^k)\| \leq \texttt{breaking\_factor} \cdot \|\widehat{y}^{k+1} - y^k\|$:

   $$\eta_{k+1} \leftarrow \widehat{\eta}_{k+1}, \quad \theta_{k+1} \leftarrow \widehat{\theta}_{k+1}, \quad \text{and} \quad y^{k+1} \leftarrow \widehat{y}^{k+1}.$$

   **Else**: reduce the step size as follows and then **go to** Step 3:

   $$\widehat{\eta}_{k+1} \leftarrow \texttt{downscaling\_factor} \cdot \widehat{\eta}_{k+1}, \quad \widehat{\theta}_{k+1} \leftarrow \frac{\eta_k}{\widehat{\eta}_{k+1}}.$$

In our experiments, we fix `breaking_factor = 1` on guidance from the authors of [43]. We then perform a grid search on `downscaling_factor` $\in \{0.4, 0.5, 0.6, 0.7, 0.8, 0.9\}$ and `interpolation_coefficient` $\in \{0.4, 0.5, 0.6, 0.7, 0.8, 0.9, 1.0\}$.

The single best configuration (by count of solved instances) for both $\epsilon = 10^{-4}$ and $\epsilon = 10^{-8}$ is `downscaling_factor` $= 0.5$ and `interpolation_coefficient` $= 0.4$.

Table 6: Performance statistics: MIP Relaxations Restarts Tol 1E-08

| Experiment | Solved count | KKT passes SGM10 | Solve time secs SGM10 |
|---|---|---|---|
| No restart | 197 | 22488.2 | 960.5 |
| Adaptive restart (theory) | 263 | 12175.6 | 308.0 |
| PDLP | 283 | 9773.3 | 215.5 |

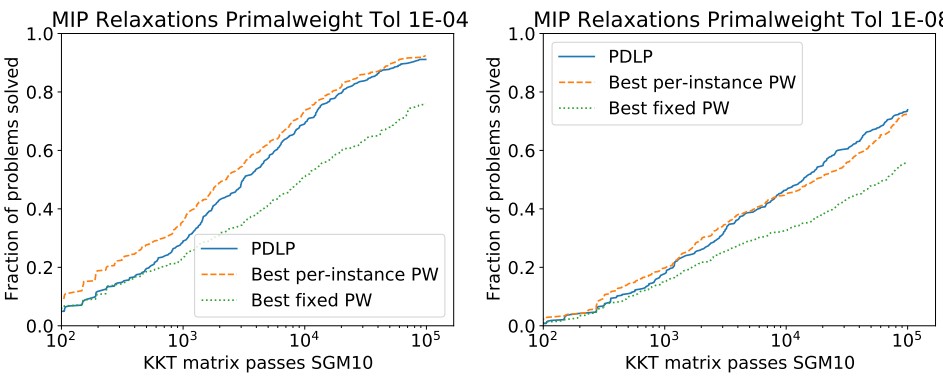

Figure 5: Primal weight ablation experiments on `MIP Relaxations`

Table 7: Performance statistics: MIP Relaxations Primalweight Tol 1E-04

| Experiment | Solved count | KKT passes SGM10 | Solve time secs SGM10 |
|---|---|---|---|
| Best fixed PW | 291 | 6548.0 | 184.4 |
| PDLP | 349 | 3058.3 | 51.5 |
| Best per-instance PW | 354 | 2091.3 | 41.2 |

## C.2 Adaptive restarts

For PDLP, we use $\beta_{\text{sufficient}} = 0.9$, $\beta_{\text{necessary}} = 0.1$, and $\beta_{\text{artificial}} = 0.5$ as the restart parameters. For "Adaptive restart (theory)" mode we match [7] and by setting $\beta_{\text{sufficient}} = \beta_{\text{necessary}} = 0.37 = \exp(-1)$. This is equivalent to removing condition (ii) from the restart criteria. For "no restart" mode we disable restarts. For this setting primal weight updates still occur when an artificial restart would have been triggered. In other words, the primal weights are updated on iteration $2, 2^2, 2^3, \ldots$.

The performance of adaptive restarts are summarized in Figure 4 and Tables 5 and 6. We can see PDLP outperforms "Adaptive restart (theory)" mode which in turn beats 'no restart" mode. This difference is much more pronounced at high accuracy.

## C.3 Primal weight updates

In PDLP, the smoothing parameter is set to $\theta = 0.5$. As baselines for PDLP's primal weight (PW) updating rule, we compare with using fixed primal weights, setting the primal weight to $\omega = \xi \cdot \texttt{InitializePrimalWeight}(c, q)$ with the bias $\xi \in \{10^{-5}, 10^{-4}, \ldots, 10^0, \ldots, 10^4, 10^5\}$ chosen by grid search. For these experiments, the smoothing parameter is set to $\theta = 0$ to fix the primal weight during the solve.

We compute both the single best value of $\xi$ (by count of solved instances), and the best per-instance value, which defines a "virtual" solver. The single best value of $\xi$ is 0.1 at both $\epsilon = 10^{-4}$ and $\epsilon = 10^{-8}$. Qualitatively, the performance of $\xi = 1$, which is a natural default, is very similar to that of $\xi = 0.1$.

From Figure 5 and Tables 7 and 8, we conclude that PDLP is competitive with the best per-instance fixed primal weight at low accuracy, and outperforms it at high accuracy.

Table 8: Performance statistics: MIP Relaxations Primalweight Tol 1E-08

| Experiment | Solved count | KKT passes SGM10 | Solve time secs SGM10 |
|---|---|---|---|
| Best fixed PW | 214 | 17852.7 | 707.2 |
| Best per-instance PW | 277 | 9846.8 | 246.0 |
| PDLP | 283 | 9773.3 | 216.8 |

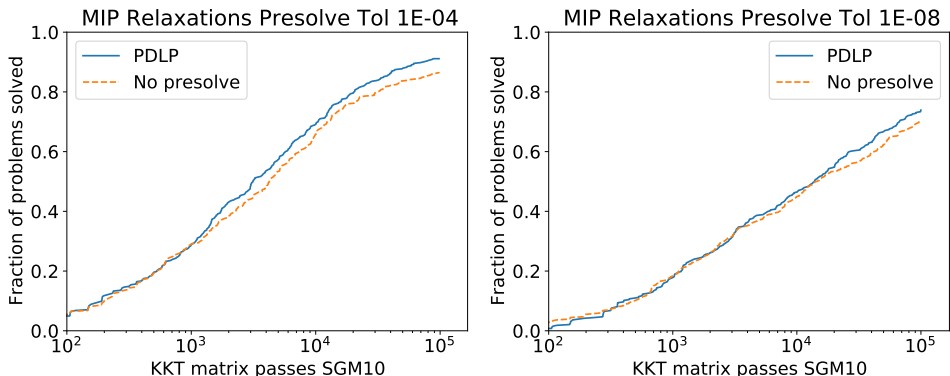

Figure 6: Presolve ablation experiments on `MIP Relaxations`

Table 9: Performance statistics: MIP Relaxations Presolve Tol 1E-04

| Experiment | Solved count | KKT passes SGM10 | Solve time secs SGM10 |
|---|---|---|---|
| No presolve | 332 | 3615.3 | 89.9 |
| PDLP | 349 | 3058.3 | 51.0 |

### C.4 Presolve

Figure 6 and Tables 9 and 10 measure the impact of presolve. Note that the impact on solve time is greater than the impact on KKT passes, because presolve also makes each KKT pass faster by making the problem smaller.

### C.5 Diagonal preconditioning

Tables 11 and 12 compare the performance of the four diagonal preconditioning techniques as mentioned in Section 3.5. As we can see, the number of solved problems of our proposed preconditioner (Ruiz and Pock-Chambolle) significantly outperform no scaling and the baselines (Pock-Chambolle or Ruiz individually).

Furthermore, Figure 7 shows the number of solved instances as a function of KKT passes for the four different diagonal preconditioners, which further shows a clear separation between PDLP and the baselines.

## D  Additional details on the PageRank LP formulation

Based on Nesterov [50], we formulate the problem of finding a maximal right eigenvector of a stochastic matrix $S$ as a feasible solution of the LP problem:

$$
\begin{aligned}
\text{find} \quad & x \\
\text{subject to:} \quad Sx \quad &\leq \quad x \\
\mathbf{1}^\top x \quad &= \quad 1 \\
x \quad &\geq \quad 0
\end{aligned}
\tag{7}
$$

Nesterov [50] states the constraint $\|x\|_\infty \geq 1$ to enforce $x \neq 0$. We instead use $\mathbf{1}^\top x = 1$ which is equivalent under scaling.

For a random scalable collection of pagerank instances, we used Barabási-Albert [9] preferential attachment graphs, using the Julia `LightGraphs.SimpleGraphs.barabasi_albert` generator

Table 10: Performance statistics: MIP Relaxations Presolve Tol 1E-08

| Experiment | Solved count | KKT passes SGM10 | Solve time secs SGM10 |
|---|---|---|---|
| No presolve | 269 | 10030.6 | 318.0 |
| PDLP | 283 | 9773.3 | 215.8 |

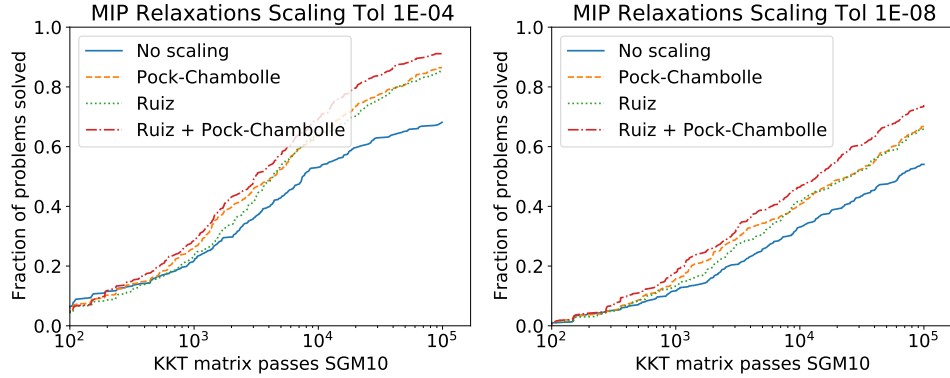

Figure 7: Diagonal preconditioning ablation experiments on `MIP Relaxations`

Table 11: Performance statistics: MIP Relaxations Scaling Tol 1E-04

| Experiment | Solved count | KKT passes SGM10 | Solve time secs SGM10 |
|---|---|---|---|
| No scaling | 261 | 6700.9 | 254.1 |
| Ruiz | 326 | 4487.7 | 88.3 |
| Pock-Chambolle | 331 | 3941.3 | 78.0 |
| Ruiz + Pock-Chambolle | 349 | 3058.3 | 50.9 |

with degree set to 3. We then computed the adjacency matrix and scaled the columns to make the matrix stochastic; call this matrix $S'$. Following the standard PageRank formulation we apply a damping factor to $S'$ and consider $S := \lambda S' + (1 - \lambda)J/n$ (where $J = \mathbf{1}\mathbf{1}^\top$ is the all-ones matrix). Intuitively, $S$ encodes a random walk that follows a link in the graph with probability $\lambda$ or jumps to a uniformly random node with probability $1 - \lambda$.

The direct approach to the damping factor results in a completely dense matrix. Instead we use the fact that $Jx = 1$ to rewrite the constraint $Sx \leq x$ in (7) as

$$\lambda(S'x)_i + (1 - \lambda)/n \ \leq \ x_i \ \forall i \ . \tag{8}$$

# E    Additional PDLP improvements results

Tables 13 and 14 give a tabular version of the impact of PDLP's improvements on the `MIP Relaxations` dataset (corresponding to Figure 1a). Tables 15 and 16 give a tabular version of the impact of PDLP's improvements on the `LP benchmark` dataset (corresponding to Figure 1b). Tables 17 and 18 give a tabular version of the impact of PDLP's improvements on the `Netlib` dataset (corresponding to Figure 1c).

# F    Additional baseline comparison results

Tables 19 and 20 give a tabular version of the comparison of PDLP with other first-order baselines on the `MIP Relaxations` dataset, Tables 21 and 22 give a tabular version of the comparison of PDLP with other first-order baselines on the `LP benchmark` dataset, and Tables 23 and 24 give a tabular version of the comparison of PDLP with other first-order baselines on the `Netlib` dataset (corresponding to Figure 2). Each of these experiments is run with a time limit of 1 hour. If the instance is unsolved, the KKT passes are set to 100,000, and the solve time to 1 hour.

Table 12: Performance statistics: MIP Relaxations Scaling Tol 1E-08

| Experiment | Solved count | KKT passes SGM10 | Solve time secs SGM10 |
|---|---|---|---|
| No scaling | 207 | 19770.0 | 787.1 |
| Ruiz | 252 | 14028.2 | 379.9 |
| Pock-Chambolle | 256 | 12960.0 | 366.2 |
| Ruiz + Pock-Chambolle | 283 | 9773.3 | 218.0 |

Table 13: Performance statistics: MIP Relaxations Improvements Tol 1E-04

| Experiment | Solved count | KKT passes SGM10 | Solve time secs SGM10 |
|---|---|---|---|
| PDHG | 113 | 38958.0 | 1088.3 |
| +restarts | 140 | 29739.6 | 770.4 |
| +scaling | 221 | 14801.5 | 313.6 |
| +primal weight | 315 | 7228.1 | 110.8 |
| +step size | 332 | 3615.3 | 67.6 |
| +presolve (= PDLP) | 349 | 3058.3 | 42.1 |

Table 14: Performance statistics: MIP Relaxations Improvements Tol 1E-08

| Experiment | Solved count | KKT passes SGM10 | Solve time secs SGM10 |
|---|---|---|---|
| PDHG | 48 | 68588.8 | 2232.6 |
| +restarts | 101 | 47301.1 | 1284.0 |
| +scaling | 162 | 25985.7 | 595.7 |
| +primal weight | 223 | 18273.4 | 331.3 |
| +step size | 269 | 10091.0 | 181.8 |
| +presolve (= PDLP) | 283 | 9773.3 | 131.5 |

Table 15: Performance statistics: LP Benchmark Improvements Tol 1E-04

| Experiment | Solved count | KKT passes SGM10 | Solve time secs SGM10 |
|---|---|---|---|
| PDHG | 10 | 64120.0 | 2148.5 |
| +restarts | 10 | 58285.8 | 2033.9 |
| +scaling | 17 | 44984.7 | 1600.3 |
| +primal weight | 37 | 22232.1 | 880.1 |
| +step size | 36 | 13003.7 | 542.5 |
| +presolve (= PDLP) | 36 | 14721.1 | 630.7 |

Table 16: Performance statistics: LP Benchmark Improvements Tol 1E-08

| Experiment | Solved count | KKT passes SGM10 | Solve time secs SGM10 |
|---|---|---|---|
| PDHG | 4 | 87478.5 | 2784.6 |
| +restarts | 7 | 69299.7 | 2210.4 |
| +scaling | 10 | 58808.8 | 1929.2 |
| +primal weight | 14 | 52872.7 | 1644.8 |
| +step size | 23 | 38630.3 | 1336.5 |
| +presolve (= PDLP) | 23 | 35106.0 | 1281.7 |

Table 17: Performance statistics: Netlib Improvements Tol 1E-04

| Experiment | Solved count | KKT passes SGM10 | Solve time secs SGM10 |
|---|---|---|---|
| PDHG | 14 | 84783.8 | 1879.2 |
| +restarts | 15 | 83879.8 | 1816.4 |
| +scaling | 43 | 55967.3 | 485.9 |
| +primal weight | 94 | 14227.5 | 30.3 |
| +step size | 98 | 9443.3 | 20.4 |
| +presolve (= PDLP) | 103 | 5405.0 | 11.8 |

Table 18: Performance statistics: Netlib Improvements Tol 1E-08

| Experiment | Solved count | KKT passes SGM10 | Solve time secs SGM10 |
|---|---|---|---|
| PDHG | 4 | 97135.7 | 2962.2 |
| +restarts | 8 | 90532.9 | 2432.7 |
| +scaling | 22 | 71722.9 | 1217.9 |
| +primal weight | 67 | 36843.2 | 167.1 |
| +step size | 85 | 23264.1 | 61.0 |
| +presolve (= PDLP) | 88 | 13419.9 | 41.2 |

Table 19: Performance statistics: MIP Relaxations Baselines Tol 1E-04

| Experiment | Solved count | KKT passes SGM10 | Solve time secs SGM10 |
|---|---|---|---|
| PDHG | 159 | 45720.5 | 922.9 |
| SCS (matrix-free) | 287 | 37027.2 | 257.0 |
| SCS | 317 | - | 149.7 |
| Enh. Extragradient | 351 | 6028.7 | 75.2 |
| PDLP | 371 | 3236.6 | 38.4 |

Table 20: Performance statistics: MIP Relaxations Baselines Tol 1E-08

| Experiment | Solved count | KKT passes SGM10 | Solve time secs SGM10 |
|---|---|---|---|
| PDHG | 80 | 79085.1 | 2026.7 |
| SCS (matrix-free) | 124 | 40486.1 | 1006.9 |
| SCS | 156 | - | 675.3 |
| Enh. Extragradient | 302 | 21216.9 | 207.4 |
| PDLP | 334 | 11381.1 | 106.4 |

Table 21: Performance statistics: LP Benchmark Baselines Tol 1E-04

| Experiment | Solved count | KKT passes SGM10 | Solve time secs SGM10 |
|---|---|---|---|
| PDHG | 12 | 67792.0 | 2009.5 |
| SCS (matrix-free) | 25 | 51040.9 | 1118.7 |
| SCS | 26 | - | 1155.6 |
| Enh. Extragradient | 27 | 25808.3 | 944.2 |
| PDLP | 32 | 16679.4 | 613.8 |

Table 22: Performance statistics: LP Benchmark Baselines Tol 1E-08

| Experiment | Solved count | KKT passes SGM10 | Solve time secs SGM10 |
|---|---|---|---|
| PDHG | 6 | 92556.5 | 2693.1 |
| SCS (matrix-free) | 7 | 63771.2 | 2155.6 |
| SCS | 9 | - | 2017.1 |
| Enh. Extragradient | 13 | 54795.9 | 1693.4 |
| PDLP | 22 | 37937.0 | 1213.4 |

Table 23: Performance statistics: Netlib Baselines Tol 1E-04

| Experiment | Solved count | KKT passes SGM10 | Solve time secs SGM10 |
|---|---|---|---|
| PDHG | 63 | 360059.0 | 558.2 |
| Enh. Extragradient | 110 | 15722.3 | 13.2 |
| SCS (matrix-free) | 110 | 26134.0 | 15.4 |
| PDLP | 113 | 6708.7 | 6.9 |
| SCS | 113 | - | 11.5 |

Table 24: Performance statistics: Netlib Baselines Tol 1E-08

| Experiment | Solved count | KKT passes SGM10 | Solve time secs SGM10 |
|---|---|---|---|
| PDHG | 44 | 391013.9 | 1376.2 |
| SCS (matrix-free) | 46 | 47943.8 | 559.1 |
| SCS | 52 | - | 345.0 |
| Enh. Extragradient | 105 | 42993.3 | 30.6 |
| PDLP | 108 | 18866.1 | 18.2 |

# G Instructions for reproducing experiments

This section documents the precise commands and command-line arguments for each experiment in the paper. These instructions are supplemental to the READMEs in the `FirstOrderLp` code directory (`https://github.com/google-research/FirstOrderLp.jl`). We assume that readers have already followed the instructions in the READMEs to set up and "instantiate" the Julia environment, and collect or generate all the datasets. Examples assume that the current working directory is `FirstOrderLp`.

The full suite of experiments takes approximately $12,500$ CPU-hours to run, and hence requires use of a cluster or cloud computing environment. Given the idiosyncrasies of these environments, we do not provide additional utilities for distributing the experiments. See Section 4.1 for details on the computing platforms we used.

The following base invocations show how to run the two main scripts without any custom arguments.

Listing 1: `solve_qp.jl` base invocation

```
julia --project=scripts scripts/solve_qp.jl
      --instance_path=$INSTANCE
      --output_dir=$OUTPUT_DIR
```

Listing 2: `solve_lp_external.jl` base invocation

```
julia --project=scripts scripts/solve_lp_external.jl
      --instance_path=$INSTANCE
      --output_dir=$OUTPUT_DIR
```

`solve_qp.jl` runs methods implemented in the `FirstOrderLp` module. `solve_lp_external.jl` runs external solvers (specifically, SCS).

## G.1 Benchmark collection

The commands used to collect the `MIP Relaxations`, `LP Benchmark`, and `Netlib` benchmarks are described in the `benchmarking` subdirectory of the `FirstOrderLp` code directory. `README.md` provides more detailed instructions, and `collect_mip_relaxations.sh`, `collect_lp_benchmark.sh`, and `collect_netlib_benchmark.sh` give illustrative scripts for collecting the benchmarks.

## G.2 Tolerances

In experiments we often solve at termination tolerances $\epsilon = 10^{-4}$ and $\epsilon = 10^{-8}$. The following command-line arguments to `solve_qp.jl` are used to set these tolerances.

Listing 3: `solve_qp.jl` arguments for $\epsilon = 10^{-4}$

```
--relative_optimality_tol 1e-4 --absolute_optimality_tol 1e-4
```

Listing 4: `solve_qp.jl` arguments for $\epsilon = 10^{-8}$

```
--relative_optimality_tol 1e-8 --absolute_optimality_tol 1e-8
```

## G.3 Improvements experiment

This section documents the command-line settings corresponding to the experiments in Section 4.2 that measure the impact of PDLP's improvements over baseline PDHG.

The following common settings apply for each run:

Listing 5: Common settings for each run

```
--kkt_matrix_pass_limit=100000
--restart_to_current_metric=gap_over_distance --verbosity=0
--method=pdhg
```

For each solve, use the base invocation for `solve_qp.jl` (Listing 1), a tolerance setting (Section G.2), common settings (Listing 5), and one set of parameters below. See the documentation in READMEs and source code for how to set $OUTPUT_DIR and process the results.

For example, the following command solves the "+ scaling" setting with $\epsilon = 10^{-4}$:

```
julia --project=scripts scripts/solve_qp.jl
      --instance_path=$INSTANCE
      --output_dir=$OUTPUT_DIR
      --relative_optimality_tol=1e-4
      --absolute_optimality_tol=1e-4
      --kkt_matrix_pass_limit=100000
      --restart_to_current_metric=gap_over_distance
      --verbosity=0 --method=pdhg
      --step_size_policy=constant
      --primal_weight_update_smoothing=0.0
      --scale_invariant_initial_primal_weight=false
```

Parameter settings:

1. "PDHG": (on original un-presolved dataset)

```
--step_size_policy=constant --l_inf_ruiz_iterations=0
--pock_chambolle_rescaling=false --l2_norm_rescaling=false
--restart_scheme=no_restart --primal_weight_update_smoothing=0.0
--scale_invariant_initial_primal_weight=false
```

2. "+ restarts":

```
--step_size_policy=constant --l_inf_ruiz_iterations=0
--pock_chambolle_rescaling=false --l2_norm_rescaling=false
--primal_weight_update_smoothing=0.0
--scale_invariant_initial_primal_weight=false
```

3. "+ scaling":

```
--step_size_policy=constant --primal_weight_update_smoothing=0.0
--scale_invariant_initial_primal_weight=false
```

4. "+primal weight":

```
--step_size_policy=constant
```

5. "+step size": No additional parameters

6. "+presolve (= PDLP)": Switch to presolved dataset.

## G.4 Comparison with other first-order baselines

This section documents the command-line settings corresponding to the experiments in Section 4.3 that compare PDLP with SCS and enhanced Extragradient.

### G.4.1 SCS (`solve_lp_external.jl`)

SCS is invoked via `solve_lp_external.jl`. The following common settings apply for all SCS runs:

Listing 6: Common settings for SCS runs

```
--scs-normalize=true --iteration_limit=1000000000
```

Because SCS does not support time limits, we use the `timeout` command to stop SCS after one hour. For example:

```
timeout 1h julia --project=scripts scripts/solve_lp_external.jl
--solver=scs-direct ...
```

The following arguments[4] are used to set $\epsilon = 10^{-4}$.

Listing 7: SCS arguments for $\epsilon = 10^{-4}$

```
--tolerance=1e-4 --scs-acceleration_lookback=0
```

The following arguments are used to set $\epsilon = 10^{-8}$.

Listing 8: SCS arguments for $\epsilon = 10^{-8}$

```
--tolerance=1e-8
```

The following arguments[5] select SCS in matrix-free mode:

Listing 9: Configuration for SCS (matrix-free)

```
--solver=scs-indirect --scs-cg_rate=1.01
```

The following arguments select SCS in its default mode that uses a cached $LDL$ factorization to solve the linear system that arises at each iteration:

Listing 10: Configuration for SCS (default)

```
--solver=scs-direct
```

### G.4.2 PDLP and Extragradient (`solve_qp.jl`)

PDLP and enhanced Extragradient are invoked via `solve_qp.jl`.

The following common settings apply to both PDLP and Extragradient.

Listing 11: Common settings for PDLP and Extragradient

```
--time_sec_limit=3600 --restart_to_current_metric=gap_over_distance
--verbosity=0
```

The following two settings select either the PDLP or enhanced Extragradient methods.

Listing 12: Configuration for PDLP

```
--method=pdhg
```

Listing 13: Configuration for enhanced Extragradient

```
--method=mirror-prox
```

### G.5 PDLP versus simplex and barrier

This section lists the commands corresponding to the experiments in Section 4.4 that compare PDLP with Gurobi's simplex and barrier algorithms.

Listing 14: Command for Gurobi Barrier

```
gurobi_cl TimeLimit=3600 Method=2 Crossover=0 Threads=1 $INSTANCE
```

Listing 15: Command for Gurobi Primal Simplex

```
gurobi_cl TimeLimit=3600 Method=0 Threads=1 $INSTANCE
```

Listing 16: Command for Gurobi Dual Simplex

```
gurobi_cl TimeLimit=3600 Method=1 Threads=1 $INSTANCE
```

---

[4]In preliminary experiments on the MIP relaxations dataset, SCS performed better at $10^{-4}$ with this custom setting of acceleration lookback, which disables Anderson Acceleration.

[5]In preliminary experiments on the MIP relaxations dataset, SCS (matrix-free) performed better with `cg_rate=1.01`, which controls the rate at which the conjugate gradient convergence tolerance decreases as a function of the iteration number.

Listing 17: Command for PDLP

```
julia --project=scripts scripts/solve_qp.jl
      --instance_path=$INSTANCE
      --output_dir=$OUTPUT_DIR
      --relative_optimality_tol=1e-8
      --absolute_optimality_tol=1e-8
      --time_sec_limit=3600
      --restart_to_current_metric=gap_over_distance
      --verbosity=0
      --method=pdhg
```

## G.6 Large-scale application: PageRank

This section describes the commands corresponding to the experiments in Section 4.5 that compares PDLP, SCS, and Gurobi's methods on PageRank instances.

The commands for Gurobi methods are the same as in Listings 14, 15, and 16. The command for PDLP is the same as in Listing 17. The command for SCS follows:

Listing 18: Command for SCS

```
timeout 1h julia --project=scripts scripts/solve_lp_external.jl
      --instance_path=$INSTANCE
      --output_dir=$OUTPUT_DIR
      --scs-normalize=true
      --iteration_limit=1000000000
      --tolerance=1e-8
      --solver=scs-indirect
      --scs-cg_rate=1.01
```

## G.7 Ablation study

In the ablation study, PDLP is invoked as:

Listing 19: PDLP configuration for the ablation study

```
  julia --project=scripts scripts/solve_qp.jl
      --instance_path=$INSTANCE
      --output_dir=$OUTPUT_DIR
      --relative_optimality_tol $TOLERANCE
      --absolute_optimality_tol $TOLERANCE
      --method=pdhg
      --restart_to_current_metric=gap_over_distance
      --kkt_matrix_pass_limit=100000
      --verbosity=0
```

on the `MIP Relaxations` dataset (to which presolve has been applied).

## G.8 Step size choice

This section describes the commands corresponding to the ablation experiments in Section C.1 on the step size choice.

The fixed step-size rule is invoked by appending the following argument to the command in Listing 19:

```
--step_size_policy=constant
```

The Malitsky and Pock step size rule is invoked by appending the following arguments to the command in Listing 19:

```
--step_size_policy=malitsky-pock
--malitsky_pock_breaking_factor=1.0
--malitsky_pock_downscaling_factor=$DOWNSCALING_FACTOR
--malitsky_pock_interpolation_coefficient=$INTERPOLATION_COEFFICIENT
```

### G.8.1 Adaptive restarts

This section describes the commands corresponding to the ablation experiments in Section C.2 on restarts.

The "No restart" setting is invoked by appending the following argument to the command in Listing 19:

```
--restart_scheme=no_restart
```

The "Adaptive restart (theory)" setting is invoked by appending the following arguments to the command in Listing 19:

```
--restart_to_current_metric=no_restart_to_current
--sufficient_reduction_for_restart=0.37
--necessary_reduction_for_restart=0.37
```

### G.8.2 Primal weight updates

This section describes the commands corresponding to the ablation experiments in Section C.3 on primal weights.

The primal weight is fixed, with the bias `$XI` $= \xi$, by appending the following arguments to the command in Listing 19:

```
--primal_weight_update_smoothing=0.0
--primal_importance=$XI
```

### G.8.3 Presolve

For the presolve ablation study in Section C.4, the "No presolve" setting is evaluated by applying PDLP to the original (non-presolved) version of the `MIP Relaxations` dataset. See `benchmarking/README.md` for more information on the dataset generation.

### G.8.4 Diagonal preconditioning

This section describes the commands corresponding to the ablation experiments in Section C.5 on diagonal preconditioning.

The "No scaling" setting corresponds to appending the following arguments to the command in Listing 19:

```
--l_inf_ruiz_iterations=0
--pock_chambolle_rescaling=false
```

The "Ruiz" setting corresponds to appending the following argument to the command in Listing 19:

```
--pock_chambolle_rescaling=false
```

The "Pock-Chambolle" setting corresponds to appending the following argument to the command in Listing 19:

```
--l_inf_ruiz_iterations=0
```

The "Ruiz + Pock-Chambolle" setting is PDLP.