# OpenReview forum: "Practical Large-Scale Linear Programming using Primal-Dual Hybrid Gradient"
_NeurIPS.cc/2021/Conference — NeurIPS 2021 Poster_

### Official Review · Reviewer_uQiX · 2021-07-09

**Rating:** 6
**Confidence:** 4

**Summary:**

The authors investigate several heuristics to accelerate the PDHG algorithm, when applied to solve large-scale linear programs.

**Main Review:**

1) My main concern is that there is no formal proof of convergence, let alone linear convergence. linear convergence of the PDHG for linear programs (and a wide range of other problems with metric subregularity) has been shown in
Alacaoglu et al. "On the convergence of stochastic primal-dual hybrid gradient". So, you should explain for each heuristic whether it keeps or not the convergence guarantees. Preconditioning preserves convergence [53], but I don't know for the other heuristics.
2) Your proposed line search is said to be faster that the one in [37], but there is no formal statement that is is valid.
3) This is a paper on convex optimization, the relationship to machine learning is not clear enough to me to justify publication in NeurIPS. A journal on optimization would be more appropriate.
4) I am not sure, but I think that the stochastic PDHG in "Stochastic Primal-Dual Hybrid Gradient Algorithm with Arbitrary Sampling and Imaging Applications" could be applied to LPs, using at every iteration activation of a random row of A or G. I would be much more convinced by such a study than by the present one, in the context of NeurIPS.

A remark: PDHG converges when tau.sigma||K||^2 <= 1, not <1, as proved in [18]. And fastest convergence is obtained in practice when tau.sigma||K||^2 = 1, so you should set eta=1/||K||.

=== update after author response period
I read all reviews and corresponding authors' responses, and I think that the authors did a very good job in responding, in a professional way, to all concerns raised.
You convinced me about the value of the paper for the practitioners willing to solve large LPs in practice. So, I revise my judgement about Neurips not being the most appropriate venue for the paper, and I recognize that the contributions are interesting for the Neurips community. Also, your reply to my point 4. is well argued. Therefore, I am increasing my score from 5 to 6. With some formal statements about convergence, I would have put an even higher score.

**Time Spent Reviewing:**

1.5

---

> ### Author Response · Authors · 2021-08-10
> **Response to Reviewer uQiX**
>
> > My main concern is that there is no formal proof of convergence, let alone linear convergence.
>
> The major contribution of this paper is experimental; we showcase that, empirically, PDHG with extensive heuristical enhancements is capable of solving large-scale linear programming problems to high accuracy. We do not claim a theoretical contribution. Putting aside the practical impact of faster methods, we believe that the optimization community is well served by such empirical comparisons, because they provide natural motivation for future theory work.
>
> > linear convergence of the PDHG for linear programs (and a wide range of other problems with metric subregularity) has been shown in Alacaoglu et al. "On the convergence of stochastic primal-dual hybrid gradient". So, you should explain for each heuristic whether it keeps or not the convergence guarantees. Preconditioning preserves convergence [53], but I don't know for the other heuristics.
>
> We agree with you about the need for further discussions on whether each heuristic can preserve theoretical guarantees. In short, on top of a basic PDHG algorithm,
> - Step size choice does not readily preserve convergence guarantees.
> - Restart criteria part (i) preserves convergence guarantees [5]. Adding part (ii) and part (iii) may break the convergence guarantees.
> - Primal weight updates do not readily preserve convergence guarantees.
> - Presolve preserves convergence guarantees.
> - Diagonal preconditioning preserves theoretical convergence guarantees.
>
> Thanks for making us aware of Alacaoglu et al. We would like to highlight that while PDHG may have linear convergence for linear programming problems, the linear convergence is sub-optimal [5]. Restarting is one way to achieve the optimal linear rate for an unconstrained bilinear problem. See more discussions on Alacaoglu et al. below.
>
> > Your proposed line search is said to be faster that the one in [37], but there is no formal statement that is is valid.
>
> The claimed improved performance of our line search rule over that of [37] is purely empirical, as we state, “our step size routine compares favorably in practice with … [37].” See Appendix C.1 for these results. We leave theoretical analysis of this line search rule for future work.
>
> > This is a paper on convex optimization, the relationship to machine learning is not clear enough to me to justify publication in NeurIPS. A journal on optimization would be more appropriate.
>
> The NeurIPS call for papers (https://nips.cc/Conferences/2021/CallForPapers) explicitly invites submissions in the area of convex optimization, in which linear programming is a fundamental problem.
>
> > I am not sure, but I think that the stochastic PDHG in "Stochastic Primal-Dual Hybrid Gradient Algorithm with Arbitrary Sampling and Imaging Applications" could be applied to LPs, using at every iteration activation of a random row of A or G. I would be much more convinced by such a study than by the present one, in the context of NeurIPS.
>
> Among a few other stochastic algorithms, we experimented with the stochastic methods presented in [Chambolle et al, 2018] (a special case of their algorithms recovers [Alacaoglu et al, 2020]). However, they worked poorly on our benchmark sets, so we decided not to present such results. There are a few fundamental reasons for why stochastic algorithms may not be suitable for linear programs:
>
> - The methods presented in [Alacaoglu et al, 2020] and [Chambolle et al, 2018] are half-stochastic, in the sense that in the primal it performs a full gradient update and in the dual it performs a stochastic gradient update (or vice versa). It is true that one can potentially do an efficient low rank update to compute the full gradient update in the primal, but such approach is no longer effective when one does a mini-batch version of stochastic gradient update, which is usually preferable in practice.
> - In practical linear programming problems (such as those we test in the paper), the constraint matrices in linear programs are usually extremely sparse, for example, having on average 10 to 100 non-zeros per row/column. As a result, sampling rows or columns of the matrix while still updating all primal/dual variables every iteration doesn't reduce runtime per iteration much. On the other hand, the progress per stochastic iteration is in general much less compared to its deterministic alternatives.
> - In practical linear programming problems, the variance in rows/columns are usually very large due to how the models are constructed. As a result, the stochastic gradient is not a good estimate of the true gradient, which is a very different case from some machine learning tasks.
>
> > A remark: PDHG converges when tau.sigma||K||^2 <= 1, not <1, as proved in [18]. And fastest convergence is obtained in practice when tau.sigma||K||^2 = 1, so you should set eta=1/||K||.
>
> We will change “<” to “<=” in a later version. The step-size choice tau.sigma||K||^2 = 1 turns out to be too conservative in practice. In order to utilize larger step-size, a more careful check in the analysis of [Theorem 1, Chambolle and Pock 2016] shows PDHG obtains sublinear rate as long as their inequality (13) is satisfied when choosing x=x^{n+1}, y=y^{n+1}, x’=x^n, y’=y^n. In other words, (13) does not need to be globally satisfied (namely tau.sigma||K||^2 <= 1), but just needs to be satisfied along the path of iterates. This analysis allows larger step-size than tau.sigma||K||^2 <= 1, and it is the foundation of our adaptive step-size rule presented in our paper. As we show in Appendix C.1 (Figure 3), there is strong evidence that our presented adaptive step-size is better than constant step-size.
>
> References:
>
> - “On the convergence of stochastic primal-dual hybrid gradient”, Alacaoglu, Fercoq and Cevher, 2020
> - “Stochastic Primal-Dual Hybrid Gradient Algorithm with Arbitrary Sampling and Imaging Applications”, Chambolle, Ehrhardt, Richtarik and Schonlieb, 2018
> - “On the ergodic convergence rates of a first-order primal–dual algorithm”, Chambolle and Pock 2016

---

### Official Review · Reviewer_eniT · 2021-07-11

**Rating:** 7
**Confidence:** 3

**Summary:**

This paper presents PDLP, a practical first-order method for linear programming (LP) that can solve to the high levels of accuracy that are expected in traditional LP applications. The proposed algorithm is matrix-free and can scale to very large problems. The core idea is based on the saddle-point reformulation of LP and the celebrated primal-dual hybrid gradient (PDHG) method (cf. Chambolle and Pock, 2011). Combined with several new techniques, e.g., diagonal preconditioning, presolving, adaptive stepsizes and adaptive restarting, PDLP improves the state-of-the-art approaches, including SCS, by a significant margin. Experimental results on PageRank applications are encouraging.


**Limitations And Societal Impact:**

1. I encourage the authors to do further literature review and include other recent methods for LP, especially FOMs.

2. It would be better to evaluate PDLP on NETLIB collections and see if it still behaves well on hard LP instances.

3. There are a couple of typos in the paper and the descriptions and language could be clearer in places.

**Main Review:**

Pros: As far as I could see, the mathematics appears to be intuitive, theoretically sound, although I did not check the details thoroughly. Despite the lack of the convergence results for several heuristics, the extensive numerical experiments were presented and supported the practical benefits of this new approach. That's the thing that I liked most about this paper since it aims at some useful approaches in real scenario. The strategies for tuning stepsizes as well as the restart criteria and diagonal preconditioning make a lot of sense to me and I appreciate that the authors put effort to explain why these techniques are useful from a practical perspective. This is an important contribution of the paper.

Cons: One of the reasons that I did not give this paper a higher score is that some relevant literatures are missing. For example, due to a very specific structure of LP, some specialized second-order algorithms exhibit superior practical performance in real applications. (also with theoretical guarantee)

X. Li, D. Sun, K. C. Toh. "An asymptotically superlinearly convergent semismooth Newton augmented Lagrangian method for Linear Programming", SIOPT, 2020.

A. Ali, E. Wong, J. Z. Kolter, "A semismooth Newton method for fast, generic convex programming", ICML 2017.

The presolving and diagonal preconditioning techniques are also combined with some first-order interior-point methods for LP. The resulting approach outperforms SCS on NETLIB collections and some other ML datasets thanks to these heuristics.

T. Lin, S. Ma, Y. Ye, S. Zhang, "An ADMM-based interior-point method for large-scale linear programming", Optimization Methods and Software, 2021.

Moreover, on account of the unique role of NETLIB collections in LP history, it would be better to evaluate PDLP on some of these datasets and see if this approach is sufficient robust in practice.

**Time Spent Reviewing:**

2 hours

---

> ### Author Response · Authors · 2021-08-10
> **Response to Reviewer eniT**
>
> > Pros: ...
>
> Thank you for highlighting the strengths and contributions of the paper.
>
> > Cons: One of the reasons that I did not give this paper a higher score is that some relevant literatures are missing. ...
>
> Thanks for bringing these references to our attention. We will cite these three papers with discussions. Specifically Li et al. and Lin et al. are prior works that combine LP presolve with FOMs; we will correct the statement in the paper that says, "We are not aware of presolve being combined with PDHG or other FOMs for LP."
>
> Li et al. use ADMM to warm-start their SNIPAL method; given PDLP’s favorable comparisons with SCS, it’s plausible that PDLP could provide a more effective warm-start. We note that their implementation didn't outperform Gurobi on any instance of MIPLIB2010. In contrast, PDLP beats Gurobi on a handful of MIPLIB2017 problems (see Table 1). Yet, we didn't find any overlap in instances between their reported results on MIPLIB2010 and ours on MIPLIB2017.
>
> In contrast with Lin et al., PDLP has a focus on solving LPs to high accuracy (i.e., up to 10^-8 relative error). Lin et al. run experiments with 10^-3 and 10^-5 relative error. Furthermore, the matrix-free/indirect methods that are computationally similar to PDLP (e.g., ABIP-CG) are compared only at 10^-3.
>
> Finally, Ali et al. consider applying a truncated semismooth Newton method to the system of equations defining a fixed point of the SCS operator. Their numerical experiments show moderate improvements over SCS for random LP problems. However, they don't present any comparisons using standard LP datasets.
>
> > Moreover, on account of the unique role of NETLIB collections in LP history, it would be better to evaluate PDLP on some of these datasets and see if this approach is sufficient robust in practice.
>
> We re-ran experiments with the 114 feasible NETLIB instances used by Lin et al. Overall, both the comparisons between methods and the impact of our heuristic improvements are qualitatively similar to those from the MIP relaxations and LP benchmark sets in our paper. PDLP can reliably solve nearly all instances in this dataset, and is substantially more robust than SCS when solving to 10^-8. We will add these results to the appendix. Details are reported below.
>
> *NETLIB instances solved at 10^-4, with a 1-hour time limit*:
> - PDLP: 113 solved
> - Enh. Extragradient: 110
> - SCS (matrix-free): 111
> - SCS: 113
>
> *NETLIB instances solved at 10^-8, with a 1-hour time limit*:
> - PDLP: 108 solved
> - Enh. Extragradient: 105
> - SCS (matrix-free): 46
> - SCS: 51
>
> *NETLIB instances solved at 10^-8, with a 100k KKT pass limit, PDLP's sequence of improvements (compare with Table 14)*:
> - (Vanilla) PDHG: 4 solved
> - Plus restarts: 8
> - Plus scaling: 22
> - Plus primal weight: 67
> - Plus step size: 85
> - Plus presolve (= PDLP): 88
>
> Note that because the NETLIB instances are relatively small, the 1-hour time limit often affords many more than 100k KKT passes, hence there’s a big difference in the number of problems solved at the 1-hour and 100k limits.
>
> > There are a couple of typos in the paper and the descriptions and language could be clearer in places.
>
> Thanks for pointing this out. We will proof-read the paper.

---

> > ### Comment · Reviewer_eniT · 2021-08-13
> > **Presolve scheme**
> >
> > Thank you for your insightful response!
> >
> > It is a bit surprising to me that SCS can achieve nice performance on 114 feasible NETLIB instances. Indeed, these instances in their original form are ill-conditioning and even have redundant rows which may fail the first-order algorithms. I have ran the experiments by myself and found that SCS performs poorly with neither presolving nor preconditioning.
> >
> > I am wondering if you apply some presolving schemes for these LP instances before you run your algorithm? If so, did you develop your own presolver or just borrow one from commercial solvers, e.g., GUROBI or CPLEX?

---

> > > ### Author Response · Authors · 2021-08-17
> > > **SCS's performance on NETLIB**
> > >
> > > > It is a bit surprising to me that SCS can achieve nice performance on 114 feasible NETLIB instances.
> > >
> > > Note that although SCS solves the same number of NETLIB instances as PDLP at 10^-4, the performance is still relatively poorer; the SGM10 (a shifted geometric mean, as defined in our paper) of the solve time in seconds is 15.2 for matrix-free SCS, 11.4 for matrix-based SCS, and 6.9 for PDLP. Also, there’s much room for improvement at 10^-8 given that SCS solves fewer than half of the instances.
> > >
> > > > I have ran the experiments by myself and found that SCS performs poorly with neither presolving nor preconditioning.
> > >
> > > > I am wondering if you apply some presolving schemes for these LP instances before you run your algorithm? If so, did you develop your own presolver or just borrow one from commercial solvers, e.g., GUROBI or CPLEX?
> > >
> > > The difference between the results we reported in our reply and your experiments may be explained by two factors: presolve and custom parameters. On the first point, we applied PaPILO, an open-source presolver, to the LP instances provided to both PDLP and SCS, for fairness, as we do in the paper. To measure the impact of presolve, we re-ran the experiments on the original, non-presolved instances. The number of instances solved dropped from 113 to 108 for matrix-based SCS and from 111 to 106 for matrix-free SCS at 10^-4. Solve times approximately doubled. Presolve indeed has an effect.
> > >
> > > Second, as noted in Appendix section G.3.1, we disable Anderson Acceleration (setting accleration_lookback to zero) at 10^-4 and set the cg_rate to 1.01 in matrix-free mode. In early testing (on other datasets) we found these settings to be beneficial for SCS. We ran an experiment on NETLIB reverting these parameters to their defaults, using also the non-presolved instances, and found a substantial degradation in performance. Matrix-free SCS solved 83 instances and matrix-based SCS solved 87 instances at 10^-4. Solve times increased by a factor of five relative to the experiment with our custom parameters and non-presolved instances.
> > >
> > > Is this performance more consistent with your experience?

---

> > > > ### Comment · Reviewer_eniT · 2021-08-21
> > > > **Thank you for your response!**
> > > >
> > > > Yes. This performance is somehow more consistent with my experience. Indeed, I also observe that Anderson Acceleration might be not helpful and run the experiment without it. However, the improvement is not as significant as your reported results. I guess this might be due to presolve scheme by PaPILO. If possible, would you like to update your code in Github such that we can rerun the experiment? This is a nice work and merits the acceptance. I have raised my score to 7.

---

> > > > > ### Author Response · Authors · 2021-08-25
> > > > > **Clarifications on SCS**
> > > > >
> > > > > Thanks for your comments!
> > > > >
> > > > > > However, the improvement is not as significant as your reported results. I guess this might be due to presolve scheme by PaPILO.
> > > > >
> > > > > To clarify, the following sets of results on NETLIB were run *without* any presolve, with a one-hour time limit, 10^-4 convergence tolerances, and SCS version 2.1.2:
> > > > >
> > > > > - AA disabled, cg_rate=1.01, all other parameters at defaults:
> > > > >   - Matrix-based SCS: 108 instances solved
> > > > >   - Matrix-free SCS: 106
> > > > > - All parameters at defaults (AA enabled, ...)
> > > > >   - Matrix-based SCS: 87 instances solved
> > > > >   - Matrix-free SCS: 83
> > > > >
> > > > > The only difference between the two runs is the SCS parameters, so presolve doesn't play a role.
> > > > >
> > > > > > If possible, would you like to update your code in Github such that we can rerun the experiment?
> > > > >
> > > > > We will update the appendix and code supplement to include instructions on how to run experiments on Netlib. Please clarify if you wish for us to post these anonymously before paper acceptance.

---

### Official Review · Reviewer_oPcr · 2021-07-15

**Rating:** 8
**Confidence:** 5

**Summary:**

The paper applies the primal-dual hybrid-gradient (PDHG) method (also sometimes referred to as Chambolle-Pock algorithm) to
solve very large linear programs. The motivation is to solve problems which are out of reach for commerical LP solvers
that are based on simplex or interior-point approaches. Moreover, the algorithm is admissible to highly parallel computation due to the simple update rules of PDHG.

The main contribution is a careful combination of existing heuristic which are known to speed up PDHG in practice: restarts, "line search" and preconditioning.

In experiments, this combination of heuristics is shown to substantially improve a basic PDHG baseline. Contrary to existing wisdom,
it is shown that FOMs, when enhanced with good heuristics, can indeed find highly accurate solutions to large-scale problems.


**Limitations And Societal Impact:**

The limitations of the approach are clearly mentioned throughout the paper; it is acknowledged that there are no convergence guarantees
and that the paper presents mainly a heuristic approach which works well in practice.

I don't believe that the paper needs to address possible negative societal impacts.


**Main Review:**

First of all, I found this paper to be very well written and enjoyable to read. The main strength of the paper is
the quality and clarity of the writing and the experimental setup. It is convincingly demonstrated
that the combination of proposed heuristics provide a strong improvement over the baseline method and existing solvers like SCS.

The introduction gives a beautiful overview and motivation of linear programming with first-order methods.
I fully agree with the points raised in the extensive and complete literature review. Also from my experience with these methods,
PDHG typically tends to perform much better than "matrix-free" implementations of ADMM or DRS.

A comprehensive comparison to other FOMs for LPs (eg. [21,28,41,63,64] mentioned in the introduction) would have been nice to see, but likely
out of scope for this work given the already very extensive experiments. Still, it would be nice to know whether PDLP is expected to perform
better than, e.g., the ECLIPSE method.

However, I see this more of a problem with the field of first-order methods of convex optimization, where countless algorithmic variants and tweaks
are proposed but not properly evaluated on challenging standard benchmarks. This paper performs a strong experimental evaluation, which can serve
as a baseline for further algorithmic developments.

In the introduction, the paper mentions that FOMs for LPs might not have a "tailing" effect due to the linear convergence rate. I found this to be
a bit misleading. To best of my knowledge, such convergence rates are all based on Hoffmann's constant which is a very loose worst case bound which
does not coincide with the practical performance at all. Rather, the fast convergence could be due to a quick identification
of the active set of constraints at the solution. Once the active set is identified, typically much faster linear convergence rates
than Hoffmann's bound apply.

After reading the paper, I was wondering why PDHG was considered, since there are many other matrix-free first order methods. Was it simply the one
that performed best in practice or is there something special about it? I feel that the fast gradient method (with restarting) on the homogeneous self-dual embedding as considered in the paper
by Neocara, Nesterov, Glineur, "Linear convergence of first order methods for non-strongly convex optimization", 2019, Sec 6.4 with similar enhancements
could also be strong candidate for linear programming.

For the experiments, the number of KKT passes is limited to 100.000 / the solve time to 1 hour. Was there a specific reason for this limitation?
Can more problems in the MIP library be solved by increasing this limit, by say, factor 10x, or are there some problems which are completely "out of reach" for first-order methods?

One might argue that the weak point of the paper is perhaps its originality, since most considered heuristics are already known.
But there are some novel ideas, for example the considered combination of Ruiz+Pock/Chambolle diagonal preconditioning, which
has a surprisingly strong effect on the performance in my opinion.

This is a matter of taste, but as mentioned, I would have liked to see a longer concluding discussion on what have we learned about
linear programming or PDHG/first-order methods from this study. Does the paper generate any new insights or shed new light on the nature of PDHG?
Does it the paper open up directions for theoreticians to work on?

For example, as shown in experiments, preconditioning of PDHG often drastically improves the
performance. But preconditioning for PDHG is far less understood than preconditioning for linear systems. For linear systems, a direct connection
of the condition number to the convergence speed can be made. But for nonlinear problems like linear-programming,
what is a good notion of "condition number"?  Can we develop a theory which guides
the design of better preconditioners, similar to the theory of preconditioning for linear systems?


**Time Spent Reviewing:**

5 hours

---

> ### Author Response · Authors · 2021-08-10
> **Response to Reviewer oPcr**
>
> > First of all, I found this paper to be very well written and enjoyable to read...
>
> Thank you for the generous characterizations of this work and for your comments.
>
> > A comprehensive comparison to other FOMs for LPs (eg. [21,28,41,63,64] mentioned in the introduction) would have been nice to see, but likely out of scope for this work given the already very extensive experiments. Still, it would be nice to know whether PDLP is expected to perform better than, e.g., the ECLIPSE method.
>
> The ECLIPSE method is targeted at LPs with special structure, and the authors solve problems to relatively low tolerances (i.e., 10^-3 primal residuals). No results on standard benchmarks are reported. Regarding other methods see our response to your paragraph: “After reading the paper …”.
>
> > In the introduction, the paper mentions that FOMs for LPs might not have a "tailing" effect due to the linear convergence rate.  I found this to be a bit misleading. To best of my knowledge, such convergence rates are all based on Hoffmann's constant which is a very loose worst case bound which does not coincide with the practical performance at all. Rather, the fast convergence could be due to a quick identification of the active set of constraints at the solution. Once the active set is identified, typically much faster linear convergence rates than Hoffmann's bound apply.
>
> We agree that the discussion about linear convergence and the tailing off effect is confusing. We will reword, “In this paper, we provide evidence that, if properly enhanced, FOMs can obtain high quality solutions to LP problems quickly. Indeed, there’s reason to expect this, as authors have developed FOMs for LP with linear rates of convergence (i.e., without a tailing-off effect) [...]. However, to our knowledge, ours is the first work to combine [...]” as
> “In this paper, we provide evidence that, if properly enhanced, FOMs can obtain high quality solutions to LP problems quickly. Indeed, there’s reason to expect this, as authors have developed FOMs for LP with linear rates of convergence [...]. On the other hand, the linear rates depend on potentially loose and hard-to-compute constants; hence, tailing off may still be observed in practice. This work addresses this gap between theory and practice by combining both theoretically-justified enhancements and practical heuristics, demonstrating their combined effectiveness with extensive computational experiments on standard benchmark instances.”
>
> Indeed, we have noticed in practice that fast linear convergence is correlated (albeit not perfectly) with identification of the final active set. Once the final active set is identified, the algorithm essentially solves a linear system. We agree that Hoffman’s constant is a loose bound.  In principle, Hoffman’s constant presents the worst case convergence rate of the algorithm for solving the corresponding linear system with an arbitrary active set. Since we do not know the final active set, it is nontrivial to derive a better linear convergence rate. This is an interesting research question for future work.
>
> > After reading the paper, I was wondering why PDHG was considered, since there are many other matrix-free first order methods.
>
> In the early stages of this project we crudely prototyped several different matrix-free methods including extragradient (which appears in this paper), stochastic PDHG (see response to uQiX), interior point methods, accelerated gradient applied to the KKT conditions similar to Neocara, Nesterov, Glineur, 2019, restarted sub-gradient descent on a non-smooth formulation [Yang and Lin, 2018], accelerated first-order methods for hyperbolic programming [Renegar 2016, 2019], an automatic restarted scheme for radial gradient descent [Renegar and Grimmer 2021, Grimmer 2018]. Of these initial prototypes, PDHG appeared to be the most promising. Of course, it may be possible that these other methods, with the right combination of additional heuristics, could also be competitive with PDLP. One of our hopes is that our work will establish new and stronger baselines against which alternative methods may be tested.
>
> > For the experiments, the number of KKT passes is limited to 100.000 / the solve time to 1 hour. Was there a specific reason for this limitation? Can more problems in the MIP library be solved by increasing this limit, by say, factor 10x, or are there some problems which are completely "out of reach" for first-order methods?
>
> The limitations of 100 000 iterations and 1 hour were primarily for convenience of running the large-scale numerical experiments. To answer your question, we ran SCS, PDLP, and Enh. Extragradient with 10-hour limits at 10^-4 and 10^-8 convergence tolerances on the set of 383 “MIP relaxations” instances. PDLP leaves only 4 instances unsolved at 10^-4 and 20 unsolved at 10^-8. The number of problems that are “out of reach” for PDLP is indeed quite small, while a substantially larger fraction remains out of reach for SCS. It would be interesting follow-up work to investigate the specific instances that remain unsolved to find an empirical cause. We summarize the new results below:
>
> *10^-4, MIP Relaxations, 10h limit (compare with table 17 w/ 1h limit):*
> - PDLP: 379 solved
> - Enh. Extragradient: 376
> - SCS (matrix-free): 346
> - SCS: 356
>
> *10^-8, MIP Relaxations, 10h limit (compare with table 18 w/ 1h limit):*
> - PDLP: 363 solved
> - Enh. Extragradient: 349
> - SCS (matrix-free): 151
> - SCS: 169
>
> > One might argue that the weak point of the paper is perhaps its originality, since most considered heuristics are already known. But there are some novel ideas, for example the considered combination of Ruiz+Pock/Chambolle diagonal preconditioning, which has a surprisingly strong effect on the performance in my opinion.
>
> We agree that many of the considered heuristics are known to the literature. Some new ideas (or new variants of algorithmic developments from the known literature) include (1) a new adaptive step-size rule, (2) a new variant of the restart scheme, (3) a new primal-weight selection rule, (4) a combination of Ruiz+Pock/Chambolle scaling (see the response to Reviewer nQvW for a more detailed discussion). As demonstrated in our numerical experiments, these new ideas turn out to have a strong effect.
>
> > This is a matter of taste, but as mentioned, I would have liked to see a longer concluding discussion on what have we learned about linear programming or PDHG/first-order methods from this study. Does the paper generate any new insights or shed new light on the nature of PDHG? Does it the paper open up directions for theoreticians to work on?
>
> Although page limitations prevented us from having a longer conclusion, we indeed hope that readers will be asking these questions. We will extend the discussion on these points in a revised version. In short, we believe the empirical part of our study showcases the effectiveness of first-order methods for solving real-life linear programming problems. We believe there’s a real potential for this approach (or similar variants) to join the standard toolkit for linear programming. On the practical side, first-order methods have many computational advantages like being able to take advantage of parallel/distributed computing, GPUs, etc. that we have not experimented with in this work. On the theoretical side, this opens up new lines of research questions. Many of our proposed heuristics call for theoretical analysis; for example, the adaptive step-size rule, and the primal weights update lack theoretical guarantees. Furthermore, beyond our proposed algorithms, we can imagine extensions like dynamic diagonal scaling, stochastic algorithms, etc. that would benefit from a foundation in theory. As you mentioned below, another related theoretical question can be what is a good notion of the “condition number” for first-order methods for linear programming and how to do preconditioning (see our response below).
>
> > Can we develop a theory which guides the design of better preconditioners, similar to the theory of preconditioning for linear systems?
>
> This is a great question. We don’t have an ultimate answer yet, but just some high-level ideas: if the active set is fixed, linear programming is equivalent to solving a linear system. When the active set changes over time, the corresponding linear system changes correspondingly. One idea is to define path-wise linear systems which naturally result in path-wise condition numbers. One could imagine dynamically preconditioning the linear system based on the current active set by taking advantage of known preconditioning techniques for linear systems.
>
> However, the theory of this approach requires additional work because the PDHG convergence analysis does not readily allow the dynamic change of norms (i.e., dynamic preconditioning). This can be an interesting research direction for future works.
>
> References:
> - “Efficient subgradient methods for general convex optimization”, Renegar, 2016
> - “Accelerated first-order methods for hyperbolic programming”, Renegar, 2019
> - “A Simple Nearly Optimal Restart Scheme For Speeding Up First-Order Methods”, Renegar and Grimmer, 2021
> - “Rsg: Beating subgradient method without smoothness and strong convexity”, Yang and Lin, 2018
> - “Radial subgradient method”, Grimmer, 2018

---

> > ### Comment · Reviewer_oPcr · 2021-08-25
> > **Thank you for the detailed and insightful response.**
> >
> > > The limitations of 100 000 iterations and 1 hour were primarily for convenience of running the large-scale numerical experiments. To answer your question, we ran SCS, PDLP, and Enh. Extragradient with 10-hour limits at 10^-4 and 10^-8 convergence tolerances on the set of 383 “MIP relaxations” instances. PDLP leaves only 4 instances unsolved at 10^-4 and 20 unsolved at 10^-8. The number of problems that are “out of reach” for PDLP is indeed quite small, while a substantially larger fraction remains out of reach for SCS. It would be interesting follow-up work to investigate the specific instances that remain unsolved to find an empirical cause. We summarize the new results below: ...
> >
> > Thank you for providing these additional runs. Is there an intuition why the 4 problems being left unsolved are particularly hard for PDLP?  For example, are they also difficult for other linear programming solvers?
> >
> > > In the early stages of this project we crudely prototyped several different matrix-free methods including extragradient (which appears in this paper), stochastic PDHG (see response to uQiX), interior point methods, accelerated gradient applied to the KKT conditions similar to Neocara, Nesterov, Glineur, 2019, restarted sub-gradient descent on a non-smooth formulation [Yang and Lin, 2018], accelerated first-order methods for hyperbolic programming [Renegar 2016, 2019], an automatic restarted scheme for radial gradient descent [Renegar and Grimmer 2021, Grimmer 2018]. Of these initial prototypes, PDHG appeared to be the most promising. Of course, it may be possible that these other methods, with the right combination of additional heuristics, could also be competitive with PDLP. One of our hopes is that our work will establish new and stronger baselines against which alternative methods may be tested.
> >
> > It is interesting to see that PDHG emerged as the most promising methods out of so many different prototypes. This is consistent with my experience with these methods. Thank you for sharing these details.
> >
> > > if the active set is fixed, linear programming is equivalent to solving a linear system. When the active set changes over time, the corresponding linear system changes correspondingly. One idea is to define path-wise linear systems which naturally result in path-wise condition numbers. One could imagine dynamically preconditioning the linear system based on the current active set by taking advantage of known preconditioning techniques for linear systems.
> >
> > I agree that this could be an interesting strategy.  This approach has been referred to as "reconditioning" (opposed to a static preconditioning at initialization) in [1] and shown to improve performance on total-variation regularized problems (however I believe without any theoretical guarantees).  Since the problem is easy once the active-set is identified, one could also have a look at the "active-set complexity", see [2].
> >
> > * [1] Ye et al., Optimization of Graph Total Variation via Active-Set-based Combinatorial Reconditioning, https://arxiv.org/abs/2002.12236
> > * [2] Nutini et al., "Active-set complexity" of proximal gradient: How long does it take to find the sparsity pattern, https://arxiv.org/abs/1712.03577

---

> > > ### Author Response · Authors · 2021-08-26
> > > **Re: four hard instances and reconditioning**
> > >
> > > > Is there an intuition why the 4 problems being left unsolved are particularly hard for PDLP? For example, are they also difficult for other linear programming solvers?
> > >
> > > The four MIP relaxation problems left unsolved are shs1014, shs1023, shs1042, and supportcase19. We noticed that the three 'shs' instances have very large artificial bounds on the variables (e.g., $x \le 10^{9}$). This impacts how PDLP computes the reduced costs by changing the set $\Lambda$ onto which we project $c - K^Ty$; that is, a reduced cost that's tiny in magnitude and happens to have the wrong sign will have a big impact on the calculated dual objective, preventing us from satisfying the convergence criteria (6a). As an experiment, we removed these large artificial variable bounds, which has the effect of counting the slightly wrong reduced costs as small dual infeasibilities rather than as contributions to the dual objective. PDLP converges to 10^-4 tolerances within 10 hours on the three modified 'shs' instances. We investigated supportcase19 but were unable to understand why PDLP converges slowly. Gurobi's barrier method is able to solve supportcase19 in under a minute.
> > >
> > > > This approach has been referred to as "reconditioning" (opposed to a static preconditioning at initialization) in [1] and shown to improve performance on total-variation regularized problems (however I believe without any theoretical guarantees). Since the problem is easy once the active-set is identified, one could also have a look at the "active-set complexity", see [2].
> > >
> > > Thanks for these relevant references! There's clearly opportunity for future work in this direction.

---

### Official Review · Reviewer_nQvW · 2021-07-17

**Rating:** 5
**Confidence:** 4

**Summary:**

This paper proposes a heuristic PDLP for large-scale LP by combining some existing techniques into PDHG. Emprical results are good, but no convergence guarantees are provided.

**Limitations And Societal Impact:**

No Societal Impact

**Main Review:**

This paper makes beneficial attempts for applying first-order methods into LP. But I found that all the tricks and heuristics in this paper are already known in existing papers --- the algorithm in this paper just simply combines these parts. As a result, this paper is more like instructions of a software package, not like an academic paper. For engineering implementation, this is a good guideline; but for academic research, this is not enough.


Some other comments:

In my understanding, a linear convergence result is not enough to prevent the tailing-off effect, because the condition number can be arbitrarily large so contracting by 1/2 is still very expensive. I believe that the reason why we say IPMs can provide high-accurate solution is they have superlinear convergence at the end of the stage which is independent of the condition number and make us get an almost exact solution in finite steps.

Algorithm 2 is the main part for the algorithm, but the name is "step size heuristic", which is not proper.

In the line 133-134, the two uses of the notation $\rho_{\cdot}^{\cdot}$ is not consistent. The first subscript is a constant, while the second one is a variable.


**Time Spent Reviewing:**

5

---

> ### Author Response · Authors · 2021-08-10
> **Response to Reviewer nQvW**
>
> > This paper makes beneficial attempts for applying first-order methods into LP. But I found that all the tricks and heuristics in this paper are already known in existing papers --- the algorithm in this paper just simply combines these parts.
>
> While many of the heuristics are inspired by other papers (as we make clear in the text), the assertion that the tricks and heuristics are already known in existing papers is not accurate:
>
> 1. The step size selection is new and tested against step size choices from literature (Appendix C1)
> 2. As described on line 134-135, 141-148. GetRestartCandidate and part (i) of the restart scheme are from [5] but all other aspects of the restart scheme, i.e., (ii)-(iii) are novel. Note that the impact of changes are tested in Figure 4 in Appendix C2 and it is clear that our changes improve over the theoretical scheme provided in [5].
> 3. The primal-weight selection is different from existing primal-weight selections [32,33] for several reasons. For example, [32,33] make very tiny changes to the primal weights at each iteration, attempting to balance the primal and dual residual. These changes have to be diminishingly small because, in our experience, PDHG may be unstable if they are too big.  In contrast, our method the primal weight is only updated during restarts, which in practice allows for much larger changes without instability issues. Moreover, our scheme tries to balance the weighted distance traveled in the primal and dual rather than the residuals. Indeed, unlike  [32,33], our scheme has been carefully designed to be scale invariant (Line 152-155).
> 4. As detailed in Section 3.5 we combine Ruiz rescaling [57] with Chambolle and Pock rescaling [53], which is original, as reviewer oPcr noted. The impact of this change is evaluated in Appendix C5.
>
> We will update the paper to make the novelty of our heuristics clearer.
>
> > As a result, this paper is more like instructions of a software package, not like an academic paper. For engineering implementation, this is a good guideline; but for academic research, this is not enough.
>
> We respectfully contest the assertion that the paper is like the instructions of a software package, and not academic research. The characterization might arguably apply if the numerical experiments are omitted from consideration. In fact, the experimental results are the main point of the paper. There is a long history of academic work whose primary contributions are algorithmic experimentation [A]. For example, some of the highest impact papers in linear programming consist of an IPM, enhanced by heuristics and tested on a benchmark set [B,C]. These papers lead to a flurry of subsequent theoretical work and proposed algorithmic tweaks, leaving us with modern interior point methods. Furthermore, many high impact papers in NeurIPS are primarily empirical [D, E].  In this paper, we advance the state of the art in FOMs and provide new baselines for future work by open-sourcing the implementation, as reviewer oPcr noted.
>
> [A] Hooker, John N. "Testing heuristics: We have it all wrong." Journal of heuristics 1.1 (1995): 33-42.
>
> [B] Mehrotra, Sanjay. "On the implementation of a primal-dual interior point method." SIAM Journal on optimization 2.4 (1992): 575-601.
>
> [C] Lustig, Irvin J., Roy E. Marsten, and David F. Shanno. "Computational experience with a primal-dual interior point method for linear programming." Linear Algebra and Its Applications 152 (1991): 191-222.
>
> [D] LeCun, Yann, et al. "Handwritten digit recognition with a back-propagation network." Advances in neural information processing systems 2 (1989).
>
> [E] Zhang, Xiang, Junbo Zhao, and Yann LeCun. "Character-level convolutional networks for text classification." Advances in neural information processing systems 28 (2015): 649-657.
>
> > In my understanding, a linear convergence result is not enough to prevent the tailing-off effect, because the condition number can be arbitrarily large so contracting by 1/2 is still very expensive.
>
> Certainly, since the linear convergence constant can be arbitrary (this statement applies to any first-order bound, for example, $\text{dist}(z^0, Z^*)$  or $L$ can be arbitrarily large depending on the problem if we are considering $O(L \text{dist}(z^0, Z^*)/t)$ rates). For this reason, it is critical to have actual experimental results verifying how well these methods work on real problems.
>
> Other reviewers noted the confusing wording in the introduction about the tailing-off effect. We will re-word this discussion as follows: “In this paper, we provide evidence that, if properly enhanced, FOMs can obtain high quality solutions to LP problems quickly. Indeed, there’s reason to expect this, as authors have developed FOMs for LP with linear rates of convergence [...]. On the other hand, the linear rates depend on potentially loose and hard-to-compute constants; hence, tailing off may still be observed in practice. This work addresses this gap between theory and practice by combining both theoretically-justified enhancements and practical heuristics, demonstrating their combined effectiveness with extensive computational experiments on standard benchmark instances.”
>
> > I believe that the reason why we say IPMs can provide high-accurate solution is they have superlinear convergence at the end of the stage which is independent of the condition number and make us get an almost exact solution in finite steps.
>
> The size of the superlinear convergence region is problem-dependent [F] and can be arbitrarily small, just as the condition number can be arbitrarily large. Consequently, it can also take arbitrarily long to reach the superlinear convergence region. In practice, interior point methods use far fewer iterations than the $\tilde{O}(\sqrt{n})$ bound from worst-case analysis [G], but exactly why is not fully understood.
>
> [F] Ye, Y., R. A. Tapia, and Y. Zhang. A Superlinearly Convergent O (sqrt {n} L)-Iteration Algorithm for Linear Programming. 1991.
>
> [G] Gondzio, Jacek. "Interior point methods 25 years later." European Journal of Operational Research 218.3 (2012): 587-601.
>
> > Algorithm 2 is the main part for the algorithm, but the name is "step size heuristic", which is not proper.
>
> To clarify, we will change the title of Algorithm 2 to “One step of PDHG using our step size heuristic”.
>
> > In the line 133-134, the two uses of the notation rho is not consistent. The first subscript is a constant, while the second one is a variable.
>
> In the second rho the radius r (which is a continuous parameter, not a constant) is set to be $|| z - z_{ref} ||_{\omega_n}$ which is a continuous value. This we believe is fairly standard notation. To clarify, we will replace “... at $z$ with radius $r$
>  [\rho definition],
> introduced by [5]” by
> “... at $z$, which for any radius $r \in (0,\infty)$ is defined by
>  [\rho definition],
> introduced by [5]”.

---

### Decision · Program_Chairs · 2021-09-27

**Decision:**

Accept (Poster)

**Comment:**

Thank you for your submission to NeurIPS. All four reviewers agree that the paper is valuable because it solves an important practical problem well. Initially, there were some concerns regarding the novelty of the methods, and the heuristic nature of the methods. Fortunately, most of these concerns were adequately addressed after discussion with the authors. Please ensure that the camera-ready version is adequately revised to capture the points raised during the discussion period, as these were important in helping the reviewers come to their final decisions.